# BiBench: Benchmarking and Analyzing Network Binarization

## Abstract

Neural network binarization is one of the most promising compression approaches with extraordinary computation and memory savings by minimizing the bit-width of weight and activation. However, despite being a general technique, recent works reveal that applying binarization in various practical scenarios, including multiple tasks, architectures, and hardware, is not trivial. Moreover, common challenges, such as severe degradation in accuracy and limited efficiency gains, suggest that specific attributes of binarization are not thoroughly studied and adequately understood. To comprehensively understand binarization methods, we present **BiBench**, a carefully engineered benchmark with in-depth analysis for network binarization. We first inspect the requirements of binarization in the actual production setting. Then for the sake of fairness and systematic, we define the evaluation tracks and metrics. We also perform a comprehensive evaluation with a rich collection of milestone binarization algorithms. Our benchmark results show that binarization still faces severe accuracy challenges, and newer state-of-the-art binarization algorithms bring diminishing improvements, even at the expense of efficiency. Moreover, the actual deployment of certain binarization operations reveals a surprisingly large deviation from their theoretical consumption. Finally, based on our benchmark results and analysis, we suggest establishing a paradigm for accurate and efficient binarization among existing techniques. We hope BiBench paves the way toward more extensive adoption of network binarization and serves as a fundamental work for future research.

## 1 Introduction

Since the rising of modern deep learning, the contradiction between ever-increasing model size and limited deployment resources has persisted. For this reason, compression technologies are crucial for practical deep learning and have been widely studied, including model quantization (Gong et al., 2014; Wu et al., 2016; Vanhoucke et al., 2011; Gupta et al., 2015), network pruning (Han et al., 2015; 2016; He et al., 2017), knowledge distillation (Hinton et al., 2015; Xu et al., 2018; Chen et al., 2018; Yim et al., 2017; Zagoruyko & Komodakis, 2017), lightweight architecture design (Howard et al., 2017; Sandler et al., 2018; Zhang et al., 2018b; Ma et al., 2018), and low-rank decomposition (Denton et al., 2014; Lebedev et al., 2015; Jaderberg et al., 2014; Lebedev & Lempitsky, 2016).

As a compression approach that extremely reduces the bit-width to 1-bit, network binarization is regarded as the most aggressive quantization technology (Rusci et al., 2020; Choukroun et al., 2019; Qin et al., 2022; Shang et al., 2022b; Zhang et al., 2022b; Bethge et al., 2020; 2019; Martinez et al., 2019; Helwegen et al., 2019). The binarized models leverage the most compact 1-bit parameters, which take little storage and memory and accelerate the inference by efficient bitwise operations. Compared to other compression technologies like network pruning and architecture design, network binarization enjoys stronger topological generics since it only applies to parameters. Therefore, in academic research, network binarization is widely studied as an independent compression technique instead of the 1-bit specialization of quantization (Gong et al., 2019; Gholami et al., 2021). It is impressive that State-of-The-Art (SoTA) binarization algorithms push binarized models to full-precision performance on large-scale tasks (Deng et al., 2009; Liu et al., 2020).

However, existing network binarization is still far from practical. We point out that two worrisome trends are emerging from accuracy and efficiency perspectives in current binarization research:

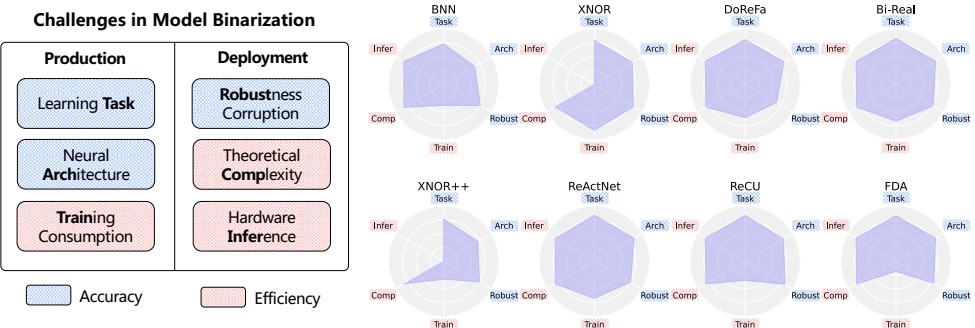

Figure 1: Evaluation tracks of BiBench. Our benchmark evaluates binarization algorithms on the most comprehensive evaluation tracks, including "Learning Task", "Neural Architecture", "Corruption Robustness", "Training Consumption", "Theoretical Complexity", and "Hardware Inference".

***Trend-1.* Accuracy comparison converging to limited scope.** In recent binarization research, several image classification tasks, *e.g.*, CIFAR-10 and ImageNet, are becoming standard options for comparing accuracy. The typical selection of evaluation tasks helps the clear and fair comparison of accuracy performance among different binarization algorithms. However, since most binarization algorithm studies are engineered for learning tasks with image modality inputs, the presented insights and conclusions are rarely verified in a broader range of other modalities and tasks. The monotonic tasks also hinder the comprehensive evaluation from an architectural perspective. Besides, data noise like corruption is a common problem on low-cost edge devices and is widely studied in compression (Lin et al., 2018; Rakin et al., 2021), whereas few advanced binarization algorithms consider the robustness of binarized models.

***Trend-2.* Efficiency analysis remaining at the theoretical level.** Network binarization is widely recognized for its significant storage and computation savings. For example, theoretical savings are up to $32\times$ and $64\times$ for convolutions, respectively (Rastegari et al., 2016; Bai et al., 2021). However, since lacking support from hardware libraries, the models compressed by binarization algorithms can hardly be evaluated on real-world edge hardware, leaving their efficiency claims lacking experimental evidence. In addition, the training efficiency of the binarization algorithm is usually neglected in current research, which causes several negative phenomena in training a binary network, such as the increasing demand for computation resources and time consumption, being sensitive to hyperparameters, and requiring detailed tuning in optimization, *etc*.

In this paper, we present **BiBench**, a network **Bi**narization **Bench**mark to evaluate binarization algorithms comprehensively from accuracy and efficiency perspectives (Table 1). Based on BiBench, we benchmark 8 representative binarization algorithms on 9 deep learning datasets, 13 different neural architectures, 2 deployment libraries, 14 hardware chips, and various hyperparameter settings. It costs us about 4 GPU years of computation time to build our BiBench, devoted to promoting comprehensive evaluation for network binarization from the perspectives of accuracy and efficiency. Furthermore, we analyze the benchmark results in depth and reveal insights along evaluation tracks, and give suggestions for designing practical binarization algorithms.

## 2 BACKGROUND

### 2.1 NETWORK BINARIZATION

Binarization compresses weights $w \in \mathbb{R}^{c_{\text{in}} \times c_{\text{out}} \times k \times k}$ and activations $a \in \mathbb{R}^{c_{\text{in}} \times w \times h}$ to 1-bit in computationally dense convolution, where $c_{\text{in}}$, $k$, $c_{\text{out}}$, $w$, and $h$ denote the input channel, kernel size, output channel, input width, and input height. The computation can be expressed as

$$o = \alpha \, \text{popcount} \left( \text{xnor} \left( \text{sign}(a), \text{sign}(w) \right) \right), \tag{1}$$

where $o$ denotes the outputs and $\alpha \in \mathbb{R}^{c_{\text{out}}}$ denotes the optional scaling factor calculated as $\alpha = \frac{\|w\|}{n}$ (Courbariaux et al., 2016b; Rastegari et al., 2016), xnor and popcount are bitwise instructions defined as (Arm, 2020; AMD, 2022). Though enjoying extreme compression and acceleration, severely limited representation causes the degradation of binarized networks. Therefore, various algorithms emerge constantly to improve accuracy (Yuan & Agaian, 2021).

The vast majority of existing binarization algorithms focus on improving the binarized operators. As shown in Eq. (1), the fundamental difference between binarized and full-precision networks is

Table 1: Comparison between BiBench and existing binarization works along evaluation tracks.

| Algorithm | Technique | | | Accurate Binarization | | | Efficient Binarization | | |
|---|---|---|---|---|---|---|---|---|---|
| | $s$ | $\tau$ | $g$ | #Task | #Arch. | Corruption Robustness | Training Consumption | Theoretical Complexity | Hardware Inference |
| BNN (Courbariaux et al., 2016b) | × | × | √ | 3 | 3 | * | √ | √ | √ |
| XNOR (Rastegari et al., 2016) | √ | × | × | 2 | 3 | * | √ | √ | √ |
| DoReFa (Zhou et al., 2016b) | √ | × | × | 2 | 2 | * | × | √ | × |
| Bi-Real (Liu et al., 2018b) | × | × | √ | 1 | 2 | × | × | √ | × |
| XNOR++ (Bulat et al., 2019) | √ | × | × | 1 | 2 | × | × | × | × |
| ReActNet (Liu et al., 2020) | × | √ | × | 1 | 2 | × | × | √ | × |
| ReCU (Xu et al., 2021b) | × | √ | √ | 2 | 4 | × | × | × | × |
| FDA (Xu et al., 2021a) | × | × | √ | 1 | 6 | × | × | × | × |
| *Our Benchmark (**BiBench**)* | √ | √ | √ | **9** | **13** | √ | √ | √ | √ |

[1] "√" and "×" indicates the track is considered in the original paper of the binarization algorithm, while "*" indicates only being studied in other related studies. "$s$", "$\tau$", or "$g$" indicates "scaling factor", "parameter redistribution", or "gradient approximation" techniques proposed in this work, respectively.

the former applies binarized operators, which also affect binarized models' optimization properties directly and determine their hardware efficiency (Alizadeh et al., 2019; Geiger & Team, 2020). In addition, the improvements of binarized operators are widely flexible across neural architectures and learning tasks (Wang et al., 2020b; Qin et al., 2022; Zhao et al., 2022), which fully exerts the generalizability of the bit-width compression. We thereby consider 8 generic binarization algorithms in our BiBench (Courbariaux et al., 2016b; Rastegari et al., 2016; Zhou et al., 2016b; Liu et al., 2018b; Bulat et al., 2019; Liu et al., 2020; Xu et al., 2021b;a). And techniques of these binarization algorithms focus on binarized operator improvement and can be categorized into several types broadly, *i.e.*, scaling factors, parameter redistribution, and gradient approximation. Note that the techniques requiring specified local structures or training pipelines are excluded for a fair comparison, such as the bi-real shortcut (Liu et al., 2018a) and duplicate activation (Liu et al., 2020). For more details of binarization algorithms for our BiBench, please see Appendix. A.

## 2.2 Challenges for Binarization

Since about 2015, network binarization has attracted great interest in different research fields, including but not limited to vision, language understanding, *etc*. However, there are still various challenges in the production and deployment process of network binarization in actual practice. The goal in production is to train accurate binarized networks with controllable resources. Recent works have revealed the capability of binarization algorithms evaluated on image classification is not always appliable to new learning tasks and neural architectures (Qin et al., 2020a; Wang et al., 2020b; Qin et al., 2021; Liu et al., 2022). And to achieve higher accuracy, some binarization algorithms require several times of training resources compared with the training of full-precision networks. Ideal binarized networks should be hardware-friendly and robust when deployed to edge devices. Unlike well-supported multi-bit (2-8 bit) quantization, most mainstream inference libraries have not yet supported the deployment of binarization on hardware (NVIDIA, 2022; HUAWEI, 2022; Qualcomm, 2022), which also causes the theoretical efficiency of existing binarization algorithms cannot align their performance on actual hardware. Moreover, the data collected by low-cost devices in natural edge scenarios is not always clean and high-quality, which affects the robustness of binarized models severely (Lin et al., 2018; Ye et al., 2019; Cygert & Czyżewski, 2021). However, recent binarization algorithms rarely consider corruption robustness when designed.

## 3 BiBench: Tracks and Metrics for Binarization

In this section, we present BiBench towards accurate and efficient network binarization. As Figure 1 shows, we build 6 evaluation tracks and corresponding metrics upon the practical challenges in production and deployment of binarization. For all tracks, higher metrics mean better performance.

### 3.1 Towards Accurate Binarization

The evaluation tracks for accurate network binarization in our BiBench are "Learning Task", "Neural Architecture" (for production), and "Corruption Robustness" (for deployment).

① **Learning Task**. We selected 9 learning tasks for 4 different data modalities to comprehensively evaluate network binarization algorithms. For the widely-evaluated 2D visual modality, in addition to image classification tasks on CIFAR-10 (Krizhevsky et al., 2014) and ImageNet (Krizhevsky et al., 2012), we also include object detection tasks on PASCAL VOC (Hoiem et al., 2009) and COCO (Lin et al., 2014) across all algorithms. For 3D visual modality, we evaluate the algorithm on ModelNet40 classification (Wu et al., 2015) and ShapeNet segmentation (Chang et al., 2015) tasks of 3D point clouds. For textual modality, the natural language understanding tasks in GLUE benchmark (Wang et al., 2018) are applied for evaluation. For speech modality, we evaluate algorithms on Speech Commands KWS task (Warden, 2018). The details of tasks and datasets are in Appendix. B.

Then we build the evaluation metric for this track. For a particular binarization algorithm, we take the accuracy of full-precision models as baselines and calculate the mean relative accuracy for all architectures on each task. Then we calculate the Overall Metric (OM) of the task track as the quadratic mean of all tasks (Curtis & Marshall, 2000). The equation of evaluation metric is

$$\text{OM}_{\text{task}} = \sqrt{\frac{1}{N}\sum_{i=1}^{N}\mathbb{E}^2\left(\frac{\boldsymbol{A}_{\text{task}_i}^{bi}}{\boldsymbol{A}_{\text{task}_i}}\right)}, \tag{2}$$

where $\boldsymbol{A}_{\text{task}_i}^{bi}$ and $\boldsymbol{A}_{\text{task}_i}$ denote the accuracy results of the binarized and full-precision models on $i$-th task, $N$ is the number of tasks, and $\mathbb{E}(\cdot)$ denote taking the mean value. The quadratic mean form is uniformly applied in BiBench to unify all metrics, which prevents metrics from being unduly influenced by certain bad items and thus can measure the overall performance on each track.

② **Neural Architecture**. We choose diverse neural architectures with mainstream CNN-based, transformer-based, and MLP-based architectures to evaluate the generalizability of binarization algorithms from the neural architecture perspective. We adopt standard ResNet-18/20/34 (He et al., 2016) and VGG (Simonyan & Zisserman, 2015) to evaluate CNN architectures, and the Faster-RCNN (Ren et al., 2015) and SSD300 (Liu et al., 2016) frameworks are applied as detectors. We binarize BERT-Tiny4/Tiny6/Base (Kenton & Toutanova, 2019) with the bi-attention mechanism for convergence (Qin et al., 2021) to evaluate transformer-based architectures. And we evaluate PointNet$_{\text{vanilla}}$ and PointNet (Qi et al., 2017) with EMA aggregator (Qin et al., 2020a), FSMN (Zhang et al., 2015), and Deep-FSMN (Zhang et al., 2018a) as typical MLP-based architectures for their linear unit composition. The details of these architectures are presented in Appendix. C.

Similar to the overall metric for learning task track, we build the metric for neural architecture:

$$\text{OM}_{\text{arch}} = \sqrt{\frac{1}{3}\left(\mathbb{E}^2\left(\frac{\boldsymbol{A}_{\text{CNN}}^{bi}}{\boldsymbol{A}_{\text{CNN}}}\right) + \mathbb{E}^2\left(\frac{\boldsymbol{A}_{\text{Transformer}}^{bi}}{\boldsymbol{A}_{\text{Transformer}}}\right) + \mathbb{E}^2\left(\frac{\boldsymbol{A}_{\text{MLP}}^{bi}}{\boldsymbol{A}_{\text{MLP}}}\right)\right)}. \tag{3}$$

③ **Corruption Robustness**. The corruption robustness of binarization on deployment is critical to deal with bad cases like perceptual device damage, a common problem with low-cost equipment in real-world implementations. We consider the robustness of binarized models to corruption of 2D visual data and evaluate algorithms on the CIFAR10-C (Hendrycks & Dietterich, 2018) benchmark.

Therefore, we evaluate binarization algorithms' performance on the corrupted data compared with the normal data using the corruption generalization gap (Zhang et al., 2022a):

$$\boldsymbol{G}_{\text{task}_i} = \boldsymbol{A}_{\text{task}_i}^{\text{norm}} - \boldsymbol{A}_{\text{task}_i}^{\text{corr}}, \tag{4}$$

where $\boldsymbol{A}_{\text{task}_i}^{\text{corr}}$ and $\boldsymbol{A}_{\text{task}_i}^{\text{norm}}$ denote the accuracy results under all architectures on $i$-th corruption task and corresponding normal task, respectively. And the overall metric on this track is calculated by

$$\text{OM}_{\text{robust}} = \sqrt{\frac{1}{C}\sum_{i=1}^{C}\mathbb{E}^2\left(\frac{\boldsymbol{G}_{\text{task}_i}}{\boldsymbol{G}_{\text{task}_i}^{bi}}\right)}. \tag{5}$$

### 3.2 TOWARDS EFFICIENT BINARIZATION

As for the efficiency of network binarization, we evaluate "Training Consumption" for production, "Theoretical Complexity" and "Hardware Inference" for deployment.

④ **Training Consumption**. We consider the occupied training resource and the hyperparameter sensitivity of binarization algorithms, which affect the consumption of one training and the whole tuning process, respectively. For each algorithm, we train its binarized networks with various hyperparameter settings, including different learning rates, learning rate schedulers, optimizers, and even random seeds, to evaluate whether the binarization algorithm is easy to tune to an optimal state. We align the epochs for binarized and full-precision networks and compare their consumption and time.

The evaluation metric for the training consumption track is related to the training time and hyperparameter sensitivity. For a specific binarization algorithm, we have

$$\mathrm{OM_{train}} = \sqrt{\frac{1}{2}\left(\mathbb{E}^2\left(\frac{\boldsymbol{T}_{\mathrm{train}}}{\boldsymbol{T}_{\mathrm{train}}^{bi}}\right) + \mathbb{E}^2\left(\frac{\mathrm{std}(\boldsymbol{A}_{\mathrm{hyper}})}{\mathrm{std}(\boldsymbol{A}_{\mathrm{hyper}}^{bi})}\right)\right)}, \tag{6}$$

where $\boldsymbol{T}_{\mathrm{train}}$ denotes the set of time in once training, $\boldsymbol{A}_{\mathrm{hyper}}$ is the set of results with different hyperparameter settings, and $\mathrm{std}(\cdot)$ is taking standard deviation values.

⑤ **Theoretical Complexity**. When evaluating theoretical complexity, we calculate the compression and speedup ratio before and after binarization on architectures such as ResNet18.

The evaluation metric relates to model size saving (MB) and computational floating-point operations (FLOPs) at inference. For binarized parameters, the storage occupation of binarized parameters is 1/32 as their 32-bit floating-point counterparts (Rastegari et al., 2016). For binarized operations, the multiplication between a 1-bit number (weight) and a 1-bit number (activation) approximately takes 1*1/64 FLOPs for a CPU with the instruction size of 64 bits (Zhou et al., 2016b; Liu et al., 2018b; Li et al., 2019). The compression ratio $\boldsymbol{r}_c$ and speedup ratio $\boldsymbol{r}_s$ are

$$\boldsymbol{r}_c = \frac{|\boldsymbol{M}|_{\ell 0}}{\frac{1}{32}\left(|\boldsymbol{M}|_{\ell 0} - |\hat{\boldsymbol{M}}|_{\ell 0}\right) + |\hat{\boldsymbol{M}}|_{\ell 0}}, \quad \boldsymbol{r}_s = \frac{\mathrm{FLOPs}_{\boldsymbol{M}}}{\frac{1}{64}\left(\mathrm{FLOPs}_{\boldsymbol{M}} - \mathrm{FLOPs}_{\hat{\boldsymbol{M}}}\right) + \mathrm{FLOPs}_{\hat{\boldsymbol{M}}}}, \tag{7}$$

where $\boldsymbol{M}$ and $\hat{\boldsymbol{M}}$ are the amount of full-precision parameters in the original and binarized models, respectively, and $\mathrm{FLOPs}_{\boldsymbol{M}}$ and $\mathrm{FLOPs}_{\hat{\boldsymbol{M}}}$ denote the computation related to these parameters, respectively. And the overall metric for theoretical complexity is

$$\mathrm{OM_{comp}} = \sqrt{\frac{1}{2}\left(\mathbb{E}^2(\boldsymbol{r}_c) + \mathbb{E}^2(\boldsymbol{r}_s)\right)}. \tag{8}$$

⑥ **Hardware Inference**. Since the limited support of binarization in hardware deployment, just two inference libraries, Larq's Compute Engine (Geiger & Team, 2020) and JD's daBNN (Zhang et al., 2019) can deploy and evaluate the binarized models on ARM hardware in practice. Regarding target hardware devices, we mainly focus on ARM CPU inference as this is the mainstream hardware type for edge scenarios, including HUAWEI Kirin, Qualcomm Snapdragon, Apple M1, MediaTek Dimensity, and Raspberry Pi. We put the hardware details in Appendix. D.

And for a certain binarization algorithm, we take the saving of storage and inference time under different inference libraries and hardware as evaluation metrics:

$$\mathrm{OM_{infer}} = \sqrt{\frac{1}{2}\left(\mathbb{E}^2\left(\frac{\boldsymbol{T}_{\mathrm{infer}}}{\boldsymbol{T}_{\mathrm{infer}}^{bi}}\right) + \mathbb{E}^2\left(\frac{\boldsymbol{S}_{\mathrm{infer}}}{\boldsymbol{S}_{\mathrm{infer}}^{bi}}\right)\right)}, \tag{9}$$

where $\boldsymbol{T}_{\mathrm{infer}}$ and $\boldsymbol{S}_{\mathrm{infer}}$ denote the inference time and storage on different devices, respectively.

## 4 BIBENCH IMPLEMENTATION

This section shows the implementation details and training and inference pipelines of our BiBench.

**Implementation details.** We implement BiBench with PyTorch (Paszke et al., 2019) packages. The definitions of the binarized operators are entirely independent of corresponding single files. And the corresponding operator in the original model can be flexibly replaced by the binarized one while evaluating different tasks and architectures. When deployed, we export well-trained binarized models of a binarization algorithm to the Open Neural Network Exchange (ONNX) format (developers, 2021) and then feed them to the inference libraries in BiBench (if it applies to this algorithm).

Table 2: Accuracy benchmark for network binarization. Blue: best in a row. Red: worst in a row.

| Track | Metric | Binarization Algorithm | | | | | | | |
|---|---|---|---|---|---|---|---|---|---|
| | | BNN | XNOR | DoReFa | Bi-Real | XNOR++ | ReActNet | ReCU | FDA |
| Learning Task (%) | CIFAR10 | 94.54 | 94.73 | 95.03 | 95.61 | 94.52 | 95.92 | 96.72 | 94.66 |
| | ImageNet | 75.81 | 77.24 | 76.61 | 78.38 | 75.01 | 78.64 | 77.98 | 78.15 |
| | VOC07 | 76.97 | 74.61 | 76.35 | 80.07 | 74.41 | 81.38 | 81.65 | 79.02 |
| | COCO17 | 77.94 | 75.37 | 78.31 | 81.62 | 79.41 | 83.82 | 85.66 | 82.35 |
| | ModelNet40 | 54.19 | 93.86 | 93.74 | 93.23 | 85.20 | 92.41 | 95.07 | 94.38 |
| | ShapeNet | 48.96 | 73.62 | 70.79 | 68.13 | 41.16 | 68.51 | 40.65 | 71.16 |
| | GLUE | 49.75 | 59.63 | 66.60 | 69.42 | 49.33 | 67.64 | 50.66 | 70.61 |
| | SpeechCom. | 75.03 | 76.93 | 76.64 | 82.42 | 68.65 | 81.86 | 76.98 | 77.90 |
| | **OM$_{task}$** | 70.82 | 78.97 | 79.82 | 81.63 | 72.89 | 81.81 | 77.96 | 81.49 |
| Neural Architecture (%) | CNNs | 72.90 | 83.74 | 83.86 | 85.02 | 78.95 | 86.20 | 83.50 | 86.34 |
| | Transformers | 49.75 | 59.63 | 66.60 | 69.42 | 49.33 | 67.64 | 50.66 | 70.61 |
| | MLPs | 64.61 | 85.40 | 85.19 | 87.83 | 76.92 | 87.13 | 86.02 | 86.14 |
| | **OM$_{arch}$** | 63.15 | 77.16 | 79.01 | 81.16 | 69.72 | 80.82 | 75.14 | 81.36 |
| Robustness Corruption (%) | CIFAR10-C | 95.26 | 100.97 | 81.43 | 96.56 | 92.69 | 94.01 | 103.29 | 98.35 |
| | **OM$_{corr}$** | 95.26 | 100.97 | 81.43 | 96.56 | 92.69 | 94.01 | 103.29 | 98.35 |

**Training and inference pipelines.** *Hype-parameters*: We train the binarized networks with the same number of epochs as their full-precision counterparts. Inspired by results in Section 5.2.1, we use the Adam optimizer for all binarized models for better convergence, the initial learning rates are $1e - 3$ as default (or $0.1\times$ of the default learning rate), and the learning rate scheduler is CosineAnnealingLR (Loshchilov & Hutter, 2017). *Architecture*: In BiBench, we thoroughly follow the original architectures of full-precision models and binarize their convolution, linear, and multiplication units with the binarization algorithms. Hardtanh is uniformly used as the activation function to avoid the all-one feature. *Pretrains*: We adopt finetuning for all binarization algorithms, and for each of them, we initialize all binarized models by the same pre-trained model for specific neural architectures and learning tasks to eliminate the inconsistency at initialization.

## 5 BIBENCH EVALUATION

This section shows our evaluation results and analysis in BiBench. The main accuracy results are in Table 2 and the efficiency results are in Table 3. More detailed results are in Appendix. E.

### 5.1 ACCURACY ANALYSIS FOR BINARIZATION

For the accuracy results of network binarization, we present the evaluation results in Table 2 for each accuracy-related track using the metrics defined in Section 3.1.

#### 5.1.1 LEARNING TASK TRACK

We present the evaluation results of binarization on various tasks. Besides the overall metrics OM$_{task}$, we also present the relative accuracy of binarized networks compared to full-precision ones.

**Accuracy retention is still the most rigorous challenge for network binarization**. With fully unified training pipelines and neural architectures, there is a large gap between the performance of binarized and full-precision networks on most learning tasks. For example, the results on large-scale ImageNet and COCO tasks are usually less than 80% of their full-precision counterparts. Moreover, the marginal effect of advanced binarization algorithms is starting to appear, *e.g.*, on the ImageNet, the SoTA algorithm ReCU is less 3% higher than vanilla BNN.

**The binarization algorithms' performances under different data modes are significantly different**. When comparing various tasks, an interesting phenomenon is that the binarized networks suffer a huge accuracy drop in language understanding GLUE benchmark, but it can almost approach full-precision performance on the ModelNet40 point cloud classification task. Similar phenomena suggest that the direct transfer of binarization studies' insights across different tasks is non-trivial.

As for the overall performance, both ReCU and ReActeNet show high accuracy across different learning tasks. Surprisingly, although ReCU wins the championship on most four individual tasks, ReActNet stands out in the overall metric comparison finally. They both apply reparameterization in the forward propagation and gradient approximation in the backward propagation.

Table 3: Efficiency benchmark for network binarization. Blue: best in a row. Red: worst in a row.

| Track | Metric | Binarization Algorithm | | | | | | | |
| | | BNN | XNOR | DoReFa | Bi-Real | XNOR++ | ReActNet | ReCU | FDA |
|---|---|---|---|---|---|---|---|---|---|
| Training Consumption (%) | Sensitivity | 27.28 | 175.53 | 113.40 | 144.59 | 28.66 | 146.06 | 53.33 | 36.62 |
| | Time Cost | 82.19 | 71.43 | 76.92 | 45.80 | 68.18 | 45.45 | 58.25 | 20.62 |
| | $OM_{train}$ | 61.23 | 134.00 | 96.89 | 107.25 | 52.30 | 108.16 | 55.84 | 29.72 |
| Theoretical Complexity (×) | Speedup | 12.60 | 12.26 | 12.37 | 12.37 | 12.26 | 12.26 | 12.37 | 12.37 |
| | Compression | 13.27 | 13.20 | 13.20 | 13.20 | 13.16 | 13.20 | 13.20 | 13.20 |
| | $OM_{comp}$ | 12.94 | 12.74 | 12.79 | 12.79 | 12.71 | 12.74 | 12.79 | 12.79 |
| Hardware Deployment (×) | Speedup | 5.45 | False | 5.45 | 5.45 | False | 4.89 | 5.45 | 5.45 |
| | Compression | 15.62 | False | 15.62 | 15.62 | False | 15.52 | 15.62 | 15.62 |
| | $OM_{infer}$ | 11.70 | False | 11.70 | 11.70 | False | 11.51 | 11.70 | 11.70 |

### 5.1.2 NEURAL ARCHITECTURE TRACK

**Binarization exhibits a clear advantage on CNN-based and MLP-based architectures over transformer-based ones**. Since being widely studied, the advanced binarization algorithms can achieve about 78%-86% of the full-precision accuracy in CNNs, and the binarized networks with MLP architectures even approach the full-precision performance (*e.g.*, Bi-Real Net 87.83%). In contrast, the transformer-based one suffers from extremely significant performance degradation when binarized, and none of the algorithms achieves an overall accuracy metric higher than 70%. Compared to CNNs and MLPs, the results show that transformer-based architectures constructed by unique attention mechanisms require specific binarization designs instead of direct binarizing.

The overall winner on the architecture track is the FDA algorithm. In this track, FDA has achieved the best results in both CNN and Transformer. The evaluation of these two tracks proves that these binarization algorithms, which apply statistical channel-wise scaling factor and custom gradient approximation like FDA and ReActNet, have the advantage of stability to a certain degree.

### 5.1.3 CORRUPTION ROBUSTNESS TRACK

**The binarized network can approach full-precision level robustness for corruption**. Surprisingly, binarized networks show robustness close to full-precision counterparts in corruption evaluation. Evaluation results on the CIFAR10-C dataset show that the binarized network performs close to the full-precision network in the typical 2D image corruption. ReCU and XNOR-Net even outperform their full-precision counterparts. If corruption robustness requirements are the same, the binarized version network requires little additional designs or supervision for robustness. Thus, binarized networks usually show comparable robustness for corruption against full-precision counterparts, which can be seen as a general property of binarized networks rather than specific algorithms.

## 5.2 EFFICIENCY ANALYSIS FOR BINARIZATION

As for efficiency, we discuss and analysis the metrics of training consumption, theoretical complexity, and hardware inference tracks below.

### 5.2.1 TRAINING CONSUMPTION TRACK

We comprehensively investigate the training cost of binarization algorithms on ResNet18 of CIFAR10 and present the sensitivity and training time results for different binarization algorithms in Table 3 and Figure 3, respectively.

**"Binarization≠Sensitivity": existing techniques can stabilize binarization-aware training**. An existing common intuition is that the training of binarized networks is usually more sensitive to the training settings than full-precision networks, caused by the representation limits and gradient approximation errors brought by the high degree of discretization. However, we find that the hyperparameter sensitivities of existing binarization algorithms are polarized. Some of them are even more hyperparameter-stable than the training of full-precision networks, while others fluctuate hugely. The reason for this problem is the difference in the techniques applied by the binarized operators of these algorithms. The training-stable binarization algorithms often have the following commonalities: (1) *Channel-wise floating-point scaling factors* based on learning or statistics; (2) *Soft gradient approximation* to reduce gradient error. These hyperparameter-stable binarization

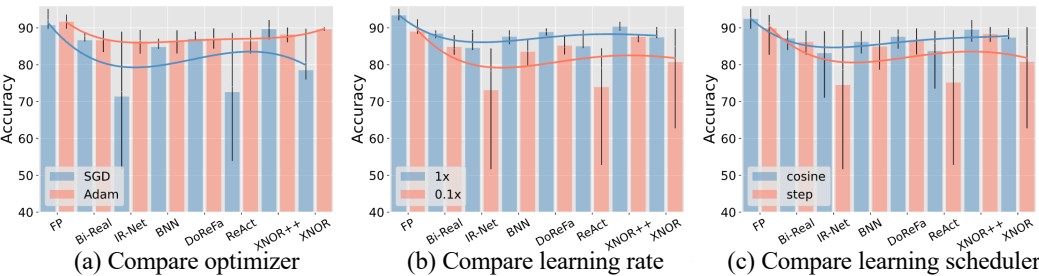

|(a) Compare optimizer | (b) Compare learning rate | (c) Compare learning scheduler|

Figure 2: Comparisons of accuracy under different training settings.

algorithms may not certainly outperform other algorithms but can simplify the tuning process in production and obtain reliable accuracy in one training.

**The preference for hyperparameter settings is evident for the training of binarization**. From the statistical results in Figure 2, training with Adam optimizer, $1\times$ (identical to full-precision network) learning rate and CosineAnnealingLR scheduler is more stable than that with other settings. Inspired by this, we adopt this setting in evaluating binarization as part of the standard training pipelines.

**The soft gradient approximation in binarization brings a significant training time increase**. Comparing the time consumed by each binarization algorithm, we found that the training time of algorithms using the custom gradient approximation techniques such as Bi-Real and ReActNet increased significantly, and the metric of FDA is even as worse as 20.62%, which means that the training time he spent is close to $5\times$ the full-precision network training.

### 5.2.2 THEORETICAL COMPLEXITY TRACK

**There is no obvious difference in the theoretical complexity among binarization algorithms.** The leading cause of the difference in compression rate is the definition of the static scaling factor of each model, *e.g.*, BNN does not apply any factors and enjoys the most compression. For theoretical acceleration, the main difference comes from two aspects. First, the static scaling factor reduction also improves the theoretical speedup. Second, real-time re-scaling and mean-shifting for activation bring additional computation, such as ReActNet, which harms $0.11\times$ speedup. In general, the theoretical complexity of each method is similar, and the overall metrics are in the range of $[12.71, 12.94]$. These results show binarization algorithms should have similar inference efficiency.

### 5.2.3 HARDWARE INFERENCE TRACK

Compared to other tracks, the hardware inference track brings us some insights for binarization in a real-world deployment.

**Limited inference libraries lead to an almost fixed paradigm of binarization deployment.** After investigating existing open-source inference libraries, we find that few inference libraries support the deployment of binarization algorithms on hardware. And there are only Larq (Geiger & Team, 2020) and daBNN (Zhang et al., 2019) with complete deployment pipelines and mainly support deployment on ARM devices. We first evaluate the

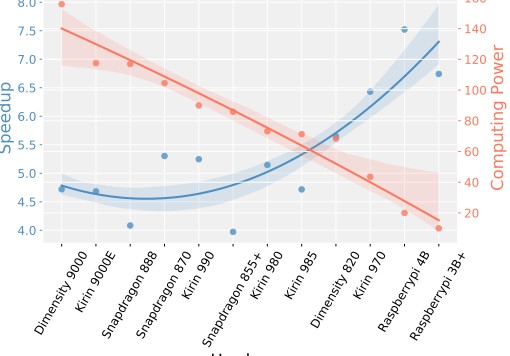

Figure 3: The lower the chip's computing power, the higher the inference speedup of deployed binarized models.

deployment capability of the two inference libraries in Table 4. Both inference libraries support a channel-wise scaling factor in floating-point form and force it to fuse into the BN layer (BN must follow every convolution of the binarized model). Neither supports dynamic activation of activations statistics nor re-scaling in inference. The only difference is that Larq further supports mean-shifting activation with a fixed bias. Constrained by the inference libraries, the practical deployment of binarization algorithms is limited. The scale factor shape of XNOR++ led to its failed deployment, and XNOR also failed because of the activation re-scaling technique. The vast majority of binarization methods have almost identical inference performance, and the mean-shifting operation of ReAct-

Table 4: Deployment capability of different inference libraries on real hardware.

| Infer. Lib. | Provider | $s$ Granularity | $s$ Form | Flod BN | Act. Re-scaling | Act. Mean-shifting |
|---|---|---|---|---|---|---|
| Larq | Larq | Channel-wise | FP32 | $\checkmark$ | $\times$ | $\checkmark$ |
| daBNN | JD | Channel-wise | FP32 | $\checkmark$ | $\times$ | $\times$ |

| Algorithm | Deployable | $s$ Granularity | $s$ Form | Flod BN | Act. Re-scaling | Act. Mean-shifting |
|---|---|---|---|---|---|---|
| BNN | $\checkmark$ | N/A | N/A | N/A | $\times$ | $\times$ |
| XNOR | $\times$ | Channel-wise | FP32 | $\checkmark$ | $\checkmark$ | $\times$ |
| DoReFa | $\checkmark$ | Channel-wise | FP32 | $\checkmark$ | $\times$ | $\times$ |
| Bi-Real | $\checkmark$ | Channel-wise | FP32 | $\checkmark$ | $\times$ | $\times$ |
| XNOR++ | $\times$ | Spatial-wise | FP32 | $\times$ | $\times$ | $\times$ |
| ReActNet | $\checkmark$ | Channel-wise | FP32 | $\checkmark$ | $\times$ | $\checkmark$ |
| ReCU | $\checkmark$ | Channel-wise | FP32 | $\checkmark$ | $\times$ | $\times$ |
| FDA | $\checkmark$ | Channel-wise | FP32 | $\checkmark$ | $\times$ | $\times$ |

Net on activation slightly affects the efficiency, *i.e.*, binarized models must satisfy fixed deployment paradigms and have almost identical efficiency performance.

**Born for the edge: the lower the chip's computing power, the higher the binarization speedup.** After deploying and evaluating binarized models across dozen chips, we compare the average speedup of the binarization algorithm on each chip. A counterintuitive finding is that the higher the chip capability, the worse the speedup of binarization on the chip (Figure 3). Further observation showed that the contradiction is mainly because higher-performance chips have more acceleration brought by multi-threading when running floating-point models. Thus the speedup of binarized models is relatively slow in these chips. The scenarios where network binarization technology comes into play better are edge chips with low performance and cost, and the extreme compression and acceleration of binarization are making deployment of advanced neural networks on edge possible.

### 5.3 Suggestions for Binarization Algorithm

Based on the above evaluation and analysis, we attempt to summarize a paradigm for accurate and efficient network binarization among existing techniques: (1) **Soft quantization approximation** is an undisputed must-have technique. This binarization technique does not affect hardware inference efficiency and is adopted by all winning binarization algorithms on accuracy tracks, including Bi-Real, ReActNet, and ReCU. (2) **Channel-wise scaling factors** are the only option available for practical binarization. The results of the learning task and neural architecture tracks demonstrate the advantage of floating-point scaling factors, and analysis of efficiency tracks further limits it to the channel-wise form. (3) **Mean-shifting the input with a fixed bias** is an optional helpful operation. Our results show that this technique in ReActNet effectively improves accuracy and consumes almost no extra computation, but not all inference libraries support it.

We have to stress that although the benchmarking on evaluation tracks leads us to several ground rules towards accurate and efficient binarization, **none of the binarization techniques or algorithms work well across all scenarios so far**. In the future, binarization research should focus on breaking the mutual restrictions between production and deployment, and the binarization algorithms should consider deployability and efficiency. The inference libraries are also expected to support more advanced binarized operators.

## 6 Discussion

In this paper, we present BiBench, a versatile and comprehensive benchmark that delves into the most fundamental questions of model binarization. BiBench covers 8 network binarization algorithms, 9 deep learning datasets (including one corruption benchmark), 13 different neural architectures, 2 deployment libraries, 14 real-world hardware, and various hyperparameter settings. Moreover, BiBench proposes evaluation tracks specifically designed to measure critical aspects such as accuracy under multiple conditions and efficiency when deployed on actual hardware. More importantly, by collating experiment results and analysis, BiBench hopes to establish an empirically optimized paradigm with several critical considerations for designing accurate and efficient binarization methods. We hope BiBench can facilitate a fair comparison of algorithms through a systematic investigation with metrics that reflect the fundamental requirements and serve as a foundation for applying model binarization in broader and more practical scenarios.

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

# APPENDIX FOR BIBENCH

## A    DETAILS OF BINARIZATION ALGORITHM

**General Binarization**:

In previous studies, quantization schemes with lower bit-widths were regarded as more aggressive schemes (Rusci et al., 2020; Choukroun et al., 2019; Qin et al., 2022), because lower bit-widths usually lead to higher compression and speed-up but result in more Severe accuracy loss. With the lowest bit-width among all quantization approaches, 1-bit quantization (binarization) is regarded as the most aggressive quantization technique (Qin et al., 2022), facing severe challenges in terms of accuracy but enjoying the highest compression and speedup ratios.

*Training*. During the training of a general binarized model, the sign function is usually applied in forward propagation, and STE or other gradient approximations is applied in backward propagation to make the binarized model trainable. Since the parameters are quantized to binary, network binarization approaches usually use a simple sign function as the quantizer instead of directly sharing the quantizer with multi-bit (2-8 bit) quantization (Gong et al., 2019; Gholami et al., 2021). Specifically, as Gong et al. (2019) describes, for multi-bit uniform quantization, given the bit width b and the floating-point activation/weight x following in the range (l, u), the complete quantization-dequantization process of uniform quantization can be defined as

$$Q_U(\boldsymbol{x}) = \text{round}\left(\frac{\boldsymbol{x}}{\Delta}\right)\Delta, \tag{10}$$

where the original range $(l, u)$ is divided into $2^b - 1$ intervals Pi, $i \in (0, 1, \cdots, 2^b - 1)$, and $\Delta = \frac{u-l}{2^b-1}$ is the interval length. When $b = 1$, the $Q_U(\boldsymbol{x})$ equals the sign function, and the binary function is expressed as

$$Q_B(\boldsymbol{x}) = \text{sign}(\boldsymbol{x}). \tag{11}$$

Therefore, binarization can be regarded as the 1-bit specialization of quantization.

*Deployment*. For real-world hardware deployment, every 32 binarized parameters are packed together using 32-bit instructions and computed simultaneously, which is the main principle for acceleration. For compression of binary algorithms. Instructions including XNOR (or combine EOR and NOT) and popcount enable binarized networks to deploy on real-world hardware. XNOR (exclusive-XOR) gate, a combination of an XOR gate followed by an inverter. XOR (also known as EOR) is a pervasive instruction that has long existed in assembly instructions for all target platforms. The popcount instruction means Population Count per byte. This instruction counts the number of bits with one value in each vector element in the source register, places the result into a vector, and writes the vector to the destination register (Arm, 2020). This instruction is applied to accelerate the inference of binarized networks (Hubara et al., 2016; Rastegari et al., 2016) and is widely supported by various hardware, *e.g.*, the definitions of popcount in ARM and x86 are in (Arm, 2020) and (AMD, 2022), respectively.

*Comparison with other compression techniques*. Most existing network compression technologies aim to reduce the size and computation of full-precision models. Specifically, knowledge distillation supervises compact small (student) models by intermediate features and/or soft outputs of the large (teacher) model (Hinton et al., 2015; Xu et al., 2018; Chen et al., 2018; Yim et al., 2017; Zagoruyko & Komodakis, 2017); model pruning (Han et al., 2015; 2016; He et al., 2017) and low-rank decomposition (Denton et al., 2014; Lebedev et al., 2015; Jaderberg et al., 2014; Lebedev & Lempitsky, 2016) reduce network parameters and computation by pruning and low-rank approximation; compact model design directly designs a compact model (Howard et al., 2017; Sandler et al., 2018; Zhang et al., 2018b; Ma et al., 2018). Although these compression technologies can effectively reduce the number of parameters, the compressed model still uses 32-bit floating-point numbers, which leaves room for further compression using model quantization/binarization technologies. Compared with multi-bit (2-8 bit) model quantization compressing parameters to integers (Gong et al., 2014; Wu et al., 2016; Vanhoucke et al., 2011; Gupta et al., 2015), binarization usually directly applies the sign function to compress the model to a more compact 1-bit (Rusci et al., 2020; Choukroun et al., 2019; Qin et al., 2022; Shang et al., 2022b; Qin et al., 2020b). Moreover, due to the application of binary parameters, bitwise operations (XNOR and popcount) can be

Table 5: The considered binarization algorithms and our final selections in BiBench. Bold means that the algorithm has an advantage in that column.

| Algorithm | Year | Conference | Citation (to 2022/11/08) | Operator Techniques | Open Source | Specified Structure / Training-pipeline |
|---|---|---|---|---|---|---|
| **BNN** (Courbariaux et al., 2016a) | 2016 | NeurIPS | **1846** | **Yes** | **Yes** | **No** |
| **XNOR-Net** (Rastegari et al., 2016) | 2016 | ECCV | **4313** | **Yes** | **Yes** | **No** |
| **DoReFa** (Zhou et al., 2016a) | 2016 | ArXiv | **174** | **Yes** | **Yes** | **No** |
| **Bi-Real Net** (Liu et al., 2018a) | 2018 | ECCV | **371** | **Yes** | **Yes** | **Optional** |
| CI-BCNN (Wang et al., 2019) | 2019 | CVPR | 77 | **Yes** | **Yes** | Yes |
| **XNOR-Net++** (Bulat et al., 2019) | 2019 | BMVC | **118** | **Yes** | **Yes** | **No** |
| RBNN (Lin et al., 2020) | 2020 | NeurIPS | 62 | **Yes** | **Yes** | **No** |
| **ReActNet** (Liu et al., 2020) | 2020 | ECCV | **151** | **Yes** | **Yes** | **Optional** |
| Si-BNN (Wang et al., 2020a) | 2020 | AAAI | 24 | **Yes** | No | **No** |
| ProxyBNN (He et al., 2020) | 2020 | ECCV | 24 | **Yes** | No | Yes |
| **FDA** (Xu et al., 2021a) | **2021** | NeurIPS | 11 | **Yes** | **Yes** | **No** |
| **ReCU** (Xu et al., 2021b) | **2021** | ICCV | 15 | **Yes** | **Yes** | **No** |
| LCR-BNN (Shang et al., 2022a) | **2022** | ECCV | 0 | **Yes** | **Yes** | Yes |

applied to inference during actual deployment instead of integer multiply-add operations in 2-8 bit model quantization. Therefore, binarization is considered to take advantage of the hardware and can achieve more speedup than multi-bit quantization.

**Selection Rules**:

First of all, we state that we obey some general rules for selecting binarization algorithms, *i.e.*, the selected binarization algorithms should improve the binarized operator since it is the fundamental difference between binarized and full-precision models (as discussed in Section 2.1). And we thus also exclude the algorithms and techniques requiring specified local structures or training pipelines for a fair comparison.

Then, we explain in detail the choice of binarization algorithms and why they are representative. When we built the BiBench, we considered various binarization algorithms with improved operator techniques in binarization research, and now we list them in detail in Table 5. We consider from the following perspectives, the purposes are making the selected binarization algorithms representative and should complete all evaluations in BiBench fairly: Operator Techniques (Yes/No), Year, Conference, Citation (to 2022/11/08), Open source (Yes/No), and Specified Structure / Training-pipeline (Yes/No/Optional).

(1) We analyze the techniques proposed in these works. Following the general rules we mentioned, all considered binarization algorithms should have significant contributions to the improvement of the binarization operator (Operator Techniques: Yes) and should not include techniques that are bound to specific architectures and training pipelines to complete well all the evaluations of the learning task, neural architecture, and training consumption tracks in BiBench (Specified Structure / Training-pipeline: No/Optional, Optional means the techniques are included but can be decoupled with binarized operator totally).

(2) We also consider the impact and reproducibility of these works. We prioritized the selection of works with more than 100 citations, which means they are more discussed and compared in binarization research and thus have higher impacts. Works in 2021 and later are regarded as the SoTA binarization algorithms and also prioritized. Furthermore, we hope the selected works have official open-source implementations for reproducibility.

Based on the above selections, eight binarization algorithms, *i.e.*, BNN, XNOR-Net, DoReFa-Net, Bi-Real Net, XNOR-Net++, ReActNet, FDA, and ReCU, stand out and are fully evaluated by our BiBench.

**BNN** (Courbariaux et al., 2016b): During the training process, BNN uses the straight-through estimator (STE) to calculate gradient $g_x$ which takes into account the saturation effect:

$$\mathtt{sign}(\boldsymbol{x}) = \begin{cases} +1, & \text{if } \boldsymbol{x} \geq 0 \\ -1, & \text{otherwise} \end{cases} \qquad \boldsymbol{g_x} = \begin{cases} \boldsymbol{g_b}, & \text{if } \boldsymbol{x} \in (-1, 1) \\ 0, & \text{otherwise.} \end{cases} \qquad (12)$$

And during inference, the computation process is expressed as

$$\boldsymbol{o} = \text{sign}(\boldsymbol{a}) \circledast \text{sign}(\boldsymbol{w}), \tag{13}$$

where $\circledast$ indicates a convolutional operation using XNOR and bitcount operations.

**XNOR-Net** (Rastegari et al., 2016): XNOR-Net obtains the channel-wise scaling factors $\boldsymbol{\alpha} = \frac{\|\boldsymbol{w}\|}{|\boldsymbol{w}|}$ for the weight and $\boldsymbol{K}$ contains scaling factors $\beta$ for all sub-tensors in activation $\boldsymbol{a}$. We can approximate the convolution between activation $\boldsymbol{a}$ and weight $\boldsymbol{w}$ mainly using binary operations:

$$\boldsymbol{o} = (\text{sign}(\boldsymbol{a}) \circledast \text{sign}(\boldsymbol{w})) \odot \boldsymbol{K}\boldsymbol{\alpha}, \tag{14}$$

where $\boldsymbol{w} \in \mathbb{R}^{c \times w \times h}$ and $\boldsymbol{a} \in \mathbb{R}^{c \times w_{\text{in}} \times h_{\text{in}}}$ denote the weight and input tensor, respectively. And the STE is also applied in the backward propagation of the training process.

**DoReFa-Net** (Zhou et al., 2016b): DoReFa-Net applies the following function for 1-bit weight and activation:

$$\boldsymbol{o} = (\text{sign}(\boldsymbol{a}) \circledast \text{sign}(\boldsymbol{w})) \odot \boldsymbol{\alpha}. \tag{15}$$

And the STE is also applied in the backward propagation with the full-precision gradient.

**Bi-Real Net** (Liu et al., 2018b): Bi-Real Net proposes a piece-wise polynomial function as the gradient approximation function:

$$\text{bireal}(\boldsymbol{a}) = \begin{cases} -1 & \text{if } \boldsymbol{a} < -1 \\ 2\boldsymbol{a} + \boldsymbol{a}^2 & \text{if } -1 \leqslant \boldsymbol{a} < 0 \\ 2\boldsymbol{a} - \boldsymbol{a}^2 & \text{if } 0 \leqslant \boldsymbol{a} < 1 \\ 1 & \text{otherwise} \end{cases}, \quad \frac{\partial \text{bireal}(\boldsymbol{a})}{\partial \boldsymbol{a}} = \begin{cases} 2 + 2\boldsymbol{a} & \text{if } -1 \leqslant \boldsymbol{a} < 0 \\ 2 - 2\boldsymbol{a} & \text{if } 0 \leqslant \boldsymbol{a} < 1 \\ 0 & \text{otherwise} \end{cases}. \tag{16}$$

And the forward propagation of Bi-Real Net is the same as Eq. (15).

**XNOR-Net++** (Bulat et al., 2019): XNOR-Net++ proposes to re-formulate Eq. (14) as:

$$\boldsymbol{o} = (\text{sign}(\boldsymbol{a}) \circledast \text{sign}(\boldsymbol{w})) \odot \boldsymbol{\Gamma}, \tag{17}$$

and we adopt the $\boldsymbol{\Gamma}$ as the following form in experiments (achieve the best performance in the original paper):

$$\boldsymbol{\Gamma} = \boldsymbol{\alpha} \otimes \boldsymbol{\beta} \otimes \boldsymbol{\gamma}, \quad \boldsymbol{\alpha} \in \mathbb{R}^{\boldsymbol{o}}, \boldsymbol{\beta} \in \mathbb{R}^{h_{\text{out}}}, \boldsymbol{\gamma} \in \mathbb{R}^{w_{\text{out}}}, \tag{18}$$

where $\boldsymbol{\alpha}$, $\boldsymbol{\beta}$, and $\boldsymbol{\gamma}$ are learnable during training.

**ReActNet** (Liu et al., 2020): ReActNet defines an RSign as a binarization function with channel-wise learnable thresholds:

$$\boldsymbol{x} = \text{rsign}(\boldsymbol{x}) = \begin{cases} +1, & \text{if } \boldsymbol{x} > \boldsymbol{\alpha} \\ -1, & \text{if } \boldsymbol{x} \leq \boldsymbol{\alpha} \end{cases}. \tag{19}$$

where $\boldsymbol{\alpha}$ is a learnable coefficient controlling the threshold. And the forward propagation is

$$\boldsymbol{o} = (\text{rsign}(\boldsymbol{a}) \circledast \text{sign}(\boldsymbol{w})) \odot \boldsymbol{\alpha}. \tag{20}$$

**ReCU** (Xu et al., 2021b): As described in their paper, ReCU is formulated as

$$\text{recu}(\boldsymbol{w}) = \max\left(\min\left(\boldsymbol{w}, Q_{(\tau)}\right), Q_{(1-\tau)}\right), \tag{21}$$

where $Q_{(\tau)}$ and $Q_{(1-\tau)}$ denote the $\tau$ quantile and $1 - \tau$ quantile of $\boldsymbol{w}$, respectively. And other implementations also strictly follow the original paper and official code.

**FDA** (Xu et al., 2021a): FDA computes the gradients of $\boldsymbol{o}$ in the backward propagation as:

$$\frac{\partial \ell}{\partial \mathbf{t}} = \frac{\partial \ell}{\partial \boldsymbol{o}} \boldsymbol{w}_2^\top \odot ((\mathbf{t}\boldsymbol{w}_1) \geq 0) \boldsymbol{w}_1^\top + \frac{\partial \ell}{\partial \boldsymbol{o}} \eta'(\mathbf{t}) + \frac{\partial \ell}{\partial \boldsymbol{o}} \odot \frac{4\omega}{\pi} \sum_{i=0}^{n} \cos((2i+1)\omega\mathbf{t}), \tag{22}$$

where $\frac{\partial \ell}{\partial \boldsymbol{o}}$ is the gradient from the upper layers, $\odot$ represents element-wise multiplication, and $\frac{\partial \ell}{\partial \mathbf{t}}$ is the partial gradient on $\mathbf{t}$ that backward propagates to the former layer. And $\boldsymbol{w}_1$ and $\boldsymbol{w}_2$ are weights in the original models and the noise adaptation modules respectively. FDA updates them as

$$\frac{\partial \ell}{\partial \boldsymbol{w}_1} = \mathbf{t}^\top \frac{\partial \ell}{\partial \boldsymbol{o}} \boldsymbol{w}_2^\top \odot ((\mathbf{t}\boldsymbol{w}_1) \geq 0), \qquad \frac{\partial \ell}{\partial \boldsymbol{w}_2} = \sigma(\mathbf{t}\boldsymbol{w}_1)^\top \frac{\partial \ell}{\partial \boldsymbol{o}}. \tag{23}$$

## B    DETAILS OF LEARNING TASKS

**Selection Rules**:

To comprehensively evaluate the performance of the binarization algorithm in various learning tasks, we should select various representative tasks. First, representative perception modalities are selected in our deep learning, including (2D/3D) vision, text, and speech. Research on these modalities has progressed rapidly and has a broad impact, so we choose specific tasks and datasets in these modalities. Specifically, (1) on the 2D vision modality, we choose the basic image classification task and object detection task (one of the most popular downstream tasks), the former including CIFAR10 and ImageNet datasets, the latter including Pascal VOC and COCO datasets. These datasets ImageNet and COCO are both more challenging large datasets, while CIFAR10 and Pascal VOC are more basic. For other modalities, binarization is still challenging even with the underlying tasks and datasets in the field, since there are few related binarization studies: (2) In the 3D vision modality, the basic point cloud classification ModelNet40 dataset is selected to evaluate the binarization performance, which is regarded as one of the most fundamental tasks in 3D point cloud research and is widely studied. (3) In the text modality, the General Language Understanding Evaluation (GLUE) benchmark is usually recognized as the most popular dataset, including nine sentence- or sentence-pair language understanding tasks. (4) In the speech modality, the keyword spotting task was chosen as the base task, specifically the Google Speech Commands classification dataset.

Based on the above reasons and rules, we have selected a series of challenging and representative tasks for BiBench to evaluate binarization comprehensively and have obtained a series of reliable and informative conclusions and experiences.

**CIFAR10** (Krizhevsky et al., 2014): The CIFAR-10 dataset (Canadian Institute For Advanced Research) is a collection of images commonly used to train machine learning and computer vision algorithms. This dataset is widely used for image classification tasks. There are 60,000 color images, each of which measures 32x32 pixels. All images are categorized into 10 different classes: airplanes, cars, birds, cats, deer, dogs, frogs, horses, ships, and trucks. Each class has 6000 images, where 5000 are for training and 1000 are for testing. The evaluation metric of the CIFAR-10 dataset is accuracy, defined as:

$$Accuracy = \frac{TP + TN}{TP + TN + FP + FN},\tag{24}$$

where *TP* (True Positive) means cases correctly identified as positive, *TN* (True Negative) means cases correctly identified as negative, *FP* (False Positive) means cases incorrectly identified as positive and *FN* (False Negative) means cases incorrectly identified as negative. To estimate the accuracy, we should calculate the proportion of *TP* and *TN* in all evaluated cases.

**ImageNet** (Krizhevsky et al., 2012): ImageNet is a dataset of over 15 million labeled high-resolution images belonging to roughly 22,000 categories. The images are collected from the web and labeled by human labelers using a crowd-sourced image labeling service called Amazon Mechanical Turk. As part of the Pascal Visual Object Challenge, ImageNet Large-Scale Visual Recognition Challenge (ILSVRC) was established in 2010. There are approximately 1.2 million training images, 50,000 validation images, and 150,000 testing images in total in ILSVRC. ILSVRC uses a subset of ImageNet, with about 1000 images in each of the 1000 categories.

ImageNet also uses accuracy to evaluate the predicted results, which is defined above.

**Pascal VOC07** (Hoiem et al., 2009): The PASCAL Visual Object Classes 2007 (VOC07) dataset contains 20 object categories including vehicles, households, animals, and other: airplane, bicycle, boat, bus, car, motorbike, train, bottle, chair, dining table, potted plant, sofa, TV/monitor, bird, cat, cow, dog, horse, sheep, and person. As a benchmark for object detection, semantic segmentation, and object classification, this dataset contains pixel-level segmentation annotations, bounding box annotations, and object class annotations. The VOC07 dataset uses mean average precision($mAP$) to evaluate results, which is defined as:

$$mAP = \frac{1}{n} \sum_{k=1}^{k=n} AP_k \tag{25}$$

where $AP_k$ denotes the average precision of the k-th category, which calculates the area under the precision-recall curve:

$$AP_k = \int_0^1 p_k(r)dr. \tag{26}$$

Especially for VOC07, we apply 11-point interpolated $AP$, which divides the recall value to $\{0.0, 0.1, \ldots, 1.0\}$ and then computes the average of maximum precision value for these 11 recall values as:

$$AP = \frac{1}{11} \sum_{r \in \{0.0,\ldots,1.0\}} AP_r \tag{27}$$

$$= \frac{1}{11} \sum_{r \in \{0.0,\ldots,1.0\}} p_{\text{interp}}r. \tag{28}$$

The maximum precision value equals to the right of its recall level:

$$p_{\text{interp}}r = \max_{\tilde{r} \geq r} p(\tilde{r}). \tag{29}$$

**COCO17** (Lin et al., 2014): The MS COCO (Microsoft Common Objects in Context) dataset is a large-scale object detection, segmentation, key-point detection, and captioning dataset. The dataset consists of 328K images. According to community feedback, in the 2017 release, the training/validation split was changed from 83K/41K to 118K/5K. And the images and annotations are the same. The 2017 test set is a subset of 41K images from the 2015 test set. Additionally, 123K images are included in the unannotated dataset. The COCO17 dataset also uses mean average precision ($mAP$) as defined above PASCAL VOC07 uses, which is defined as above.

**ModelNet40** (Wu et al., 2015): The ModelNet40 dataset contains point clouds of synthetic objects. As the most widely used benchmark for point cloud analysis, ModelNet40 is popular due to the diversity of categories, clean shapes, and well-constructed dataset. In the original ModelNet40, 12,311 CAD-generated meshes are divided into 40 categories, where 9,843 are for training, and 2,468 are for testing. The point cloud data points are sampled by a uniform sampling method from mesh surfaces and then scaled into a unit sphere by moving to the origin. The ModelNet40 dataset also uses accuracy as the metric, which has been defined above in CIFAR10.

**ShapeNet** (Chang et al., 2015): ShapeNet is a large-scale repository for 3D CAD models developed by researchers from Stanford University, Princeton University, and the Toyota Technological Institute in Chicago, USA.

Using WordNet hypernym-hyponym relationships, the repository contains over 300M models, with 220,000 classified into 3,135 classes. There are 31,693 meshes in the ShapeNet Parts subset, divided into 16 categories of objects (*i.e.*, tables, chairs, planes, *etc.*). Each shape contains 2-5 parts (with 50 part classes in total).

**GLUE** (Wang et al., 2018): General Language Understanding Evaluation (GLUE) benchmark is a collection of nine natural language understanding tasks, including single-sentence tasks CoLA and SST-2, similarity and paraphrasing tasks MRPC, STS-B and QQP, and natural language inference tasks MNLI, QNLI, RTE, and WNLI. Among them, SST-2, MRPC, QQP, MNLI, QNLI, RTE, and WNLI use accuracy as the metric, which is defined in CIFAR10. CoLA is measured by Matthews Correlation Coefficient (MCC), which is better in binary classification since the number of positive and negative samples are extremely unbalanced:

$$MCC = \frac{TP \times TN - FP \times FN}{\sqrt{(TP + FP)(TP + FN)(TN + FP)(TN + FN)}}. \tag{30}$$

And STS-B is measured by Pearson/Spearman Correlation Coefficient:

$$r_{Pearson} = \frac{1}{n-1} \sum_{i=1}^n \left(\frac{X_i - \bar{X}}{s_X}\right)\left(\frac{Y_i - \bar{Y}}{s_Y}\right), r_{Spearman} = 1 - \frac{6 \sum d_i^2}{n(n^2 - 1)}, \tag{31}$$

where $n$ is the number of observations, $s_X$ and $s_Y$ indicate the sum of squares of $X$ and $Y$ respectively, and $d_i$ is the difference between the ranks of corresponding variables.

**SpeechCom.** (Warden, 2018): As part of its training and evaluation process, SpeechCom provides a collection of audio recordings containing spoken words. Its primary goal is to provide a way to build and test small models that detect a single word that belongs to a set of ten target words. Models should detect as few false positives as possible from background noise or unrelated speech while providing as few false positives as possible. The accuracy metric for SpeechCom is also the same as CIFAR10.

**CIFAR10-C** (Hendrycks & Dietterich, 2018): CIFAR10-C is a dataset generated by adding 15 common corruptions and 4 extra corruptions to the test images in the Cifar10 dataset. It benchmarks the frailty of classifiers under corruption, including noise, blur, weather, and digital influence. And each type of corruption has five levels of severity, resulting in 75 distinct corruptions. We report the accuracy of the classifiers under each level of severity and each corruption. Meanwhile, we use the mean and relative corruption error as metrics. Denote the error rate of *Network* under *Settings* as $E_{Settings}^{Network}$. The classifier's aggregate performance across the five severities of the corruption types. The Corruption Errors of a certain type of *Corruption* is computed with the formula:

$$CE_{Corruption}^{Network} = \sum_{s=1}^{5} E_{s,Corruption}^{Network} / \sum_{s=1}^{5} E_{s,Corruption}^{AlexNet}. \tag{32}$$

To make Corruption Errors comparable across corruption types, the difficulty is usually adjusted by dividing by AlexNet's errors.

## C   DETAILS OF NEURAL ARCHITECTURES

**ResNet** (He et al., 2016): Residual Networks, or ResNets, learn residual functions concerning the layer inputs instead of learning unreferenced functions. Instead of making stacked layers directly fit a desired underlying mapping, residual nets let these layers fit a residual mapping. There is empirical evidence that these networks are easier to optimize and can achieve higher accuracy with considerably increased depth.

**VGG** (Simonyan & Zisserman, 2015): VGG is a classical convolutional neural network architecture. It is proposed by an analysis of how to increase the depth of such networks. It is characterized by its simplicity: the network utilizes small $3\times3$ filters, and the only other components are pooling layers and a fully connected layer.

**MobileNetV2** (Sandler et al., 2018): MobileNetV2 is a convolutional neural network architecture that performs well on mobile devices. This model has an inverted residual structure with residual connections between the bottleneck layers. The intermediate expansion layer employs lightweight depthwise convolutions to filter features as a source of nonlinearity. In MobileNetV2, the architecture begins with an initial layer of 32 convolution filters, followed by 19 residual bottleneck layers.

**Faster-RCNN** (Ren et al., 2015): Faster R-CNN is an object detection model that improves Fast R-CNN by utilizing a region proposal network (RPN) with the CNN model. The RPN shares full-image convolutional features with the detection network, enabling nearly cost-free region proposals. A fully convolutional network is used to predict the bounds and objectness scores of objects at each position simultaneously. RPNs use end-to-end training to produce region proposals of high quality and instruct the unified network where to search. Sharing their convolutional features allows RPN and Fast R-CNN to be combined into a single network. Faster R-CNN consists of two modules. The first module is a deep, fully convolutional network that proposes regions, and the second is the detector that uses the proposals for giving the final prediction boxes.

**SSD** (Liu et al., 2016): SSD is a single-stage object detection method that discretizes the output space of bounding boxes into a set of default boxes over different aspect ratios and scales per feature map location. During prediction, each default box is adjusted to match better the shape of the object based on its scores for each object category. In addition, the network automatically handles objects of different sizes by combining predictions from multiple feature maps with different resolutions.

**BERT** (Kenton & Toutanova, 2019): BERT, or Bidirectional Encoder Representations from Transformers, improves upon standard Transformers by removing the unidirectionality constraint using a masked language model (MLM) pre-training objective. By masking some tokens from the input, the masked language model attempts to estimate the original vocabulary id of the masked word based

solely on its context. An MLM objective differs from a left-to-right language model in that it enables the representation to integrate the left and right contexts, which facilitates pre-training a deep bidirectional Transformer. Additionally, BERT uses a next-sentence prediction task that pre-trains text-pair representations along with the masked language model. Note that we replace the direct binarized attention with a bi-attention mechanism to prevent the model from completely crashing (Qin et al., 2021).

**PointNet** (Qi et al., 2017): PointNet is a unified architecture for applications ranging from object classification and part segmentation to scene semantic parsing. The architecture directly receives point clouds as input and outputs either class labels for the entire input or point segment/part labels. **PointNet-Vanilla** is a variant of PointNet, which drops off the T-Net module. And for all PointNet models, we apply the EMA-Max (Qin et al., 2020a) as the aggregator, because directly following the max pooling aggregator will cause the binarized PointNets to fail to converge.

**FSMN** (Zhang et al., 2015): Feedforward sequential memory networks or FSMN is a novel neural network structure to model long-term dependency in time series without using recurrent feedback. It is a standard fully connected feedforward neural network containing some learnable memory blocks. As a short-term memory mechanism, the memory blocks encode long context information using a tapped-delay line structure.

**Deep-FSMN** (Zhang et al., 2018a): The Deep-FSMN architecture is an improved feedforward sequential memory network (FSMN) with skip connections between memory blocks in adjacent layers. By utilizing skip connections, information can be transferred across layers, and thus the gradient vanishing problem can be avoided when building very deep structures.

## D    DETAILS OF HARDWARE

**Hisilicon Kirin** (Hisilicon, 2022): Kirin is a series of ARM-based systems-on-a-chip (SoCs) produced by HiSilicon. Their products include Kirin 970, Kirin 980, Kirin 985, *etc.*

**MediaTek Dimensity** (MediaTek, 2022): Dimensity is a series of ARM-based systems-on-a-chip (SoCs) produced by MediaTek. Their products include Dimensity 820, Dimensity 8100, Dimensity 9000, *etc.*

**Qualcomm Snapdragon** (Singh & Jain, 2014): Snapdragon is a family of mobile systems-on-a-chip (SoC) processor architecture provided by Qualcomm. The original Snapdragon chip, the Scorpio, was similar to the ARM Cortex-A8 core based upon the ARMv7 instruction set, but it was enhanced by the use of SIMD operations, which provided higher performance. Qualcomm Snapdragon processors are based on the Krait architecture. They are equipped with an integrated LTE modem, providing seamless connectivity across 2G and 3G LTE networks.

**Raspberrypi** (Wikipedia, 2022b): Raspberry Pi is a series of small single-board computers (SBCs) developed in the United Kingdom by the Raspberry Pi Foundation in association with Broadcom. Raspberry Pi was originally designed to promote the teaching of basic computer science in schools and in developing countries. As a result of its low cost, modularity, and open design, it is used in many applications, including weather monitoring, and is sold outside the intended market. It is typically used by computer hobbyists and electronic enthusiasts due to the adoption of HDMI and USB standards.

**Apple M1** (Wikipedia, 2022a): Apple M1 is a series of ARM-based systems-on-a-chip (SoCs) designed by Apple Inc. As a central processing unit (CPU) and graphics processing unit (GPU) for Macintosh desktops and notebooks, as well as iPad Pro and iPad Air tablets. In November 2020, Apple introduced the M1 chip, followed by the professional-oriented M1 Pro and M1 Max chips in 2021. Apple launched the M1 Ultra in 2022, which combines two M1 Max chips in a single package. The M1 Max is a higher-performance version of the M1 Pro, with larger GPU cores and memory bandwidth.

## E    FULL RESULTS

Table 6-7 shows the accuracy of different binarization algorithms on 2D and 3D vision tasks, including CIFAR10, ImageNet, PASCAL VOC07, COCO14 for 2D vision tasks and ModelNet40 for

Table 6: Accuracy on 2D and 3D Vision Tasks.

| Task | Arch. | FP32 | Binarization Algorithm | | | | | | | |
| | | | BNN | XNOR | DoReFa | Bi-Real | XNOR++ | ReActNet | ReCU | FDA |
|---|---|---|---|---|---|---|---|---|---|---|
| CIFAR10 | ResNet20 | 91.99 | 85.31 | 85.53 | 85.18 | 85.56 | 85.41 | 86.18 | 86.42 | 86.38 |
| | ResNet18 | 94.82 | 89.69 | 91.4 | 91.55 | 91.20 | 90.04 | 91.55 | 92.79 | 90.42 |
| | ResNet34 | 95.34 | 90.82 | 89.58 | 90.95 | 92.50 | 90.59 | 92.69 | 93.64 | 89.59 |
| | VGG-Small | 93.80 | 89.66 | 89.65 | 89.66 | 90.25 | 89.34 | 90.27 | 90.84 | 89.48 |
| ImageNet | ResNet18 | 69.90 | 52.99 | 53.99 | 53.55 | 54.79 | 52.43 | 54.97 | 54.51 | 54.63 |
| VOC07 | Faster-RCNN | 76.06 | 58.54 | 56.75 | 58.07 | 60.90 | 56.60 | 61.90 | 62.10 | 60.10 |
| | SSD300 | 77.34 | 9.09 | 33.72 | 30.70 | 31.90 | 9.41 | 38.41 | 9.80 | 43.68 |
| COCO14 | Faster-RCNN | 27.20 | 21.20 | 20.50 | 21.30 | 22.20 | 21.60 | 22.80 | 23.30 | 22.40 |
| ModelNet40 | PointNet$_{vanilla}$ | 86.80 | 85.13 | 83.47 | 85.21 | 85.37 | 85.66 | 85.13 | 85.21 | 85.49 |
| | PointNet | 88.20 | 9.08 | 80.75 | 78.77 | 77.71 | 63.25 | 76.50 | 81.12 | 79.62 |

3D vision task. And for each task, we cover several representative model architectures and binarize them with the binarization algorithms mentioned above.

We also evaluate binarization algorithms on language and speech tasks, for which we test TinyBERT (4 layers and 6 layers) on GLUE Benchmark and FSMN and D-FSMN on the SpeechCommand dataset. Results are listed in Table 8.

To demonstrate the robustness corruption of binarized algorithms, we show the results on the CIFAR10-C benchmark, which is used to benchmark the robustness to common perturbations in Table 9 and Table 10. It includes 15 kinds of noise, blur, weather, and digital corruption, each with five levels of severity.

The sensitivity of hyperparameters while training is shown in Table 11-12. For each binarization algorithm, we use SGD or Adam optimizer, $1\times$ or $0.1\times$ of the original learning rate, cosine or step learning scheduler, and 200 training epochs. Each case is tested five times to show the training stability. We also calculate the mean and standard deviation (std) of accuracy. The best accuracy and the lowest std for each binarization algorithm are bolded.

We conduct comprehensive deployment and inference on various kinds of hardware, including the Kirin series (970, 980, 985, 990, and 9000E), Dimensity series (820 and 9000), Snapdragon series (855+, 870 and 888), Raspberrypi (3B+ and 4B) and Apple M1 series (M1 and M1 Max). Limited to the support of frameworks, we can only test BNN and ReAct with Larq compute engine and only BNN with daBNN. We convert models to enable the actual inference on real hardware, including ResNet18/34 and VGG-Small on Larq, and only ResNet18/34 on daBNN. And we test 1, 2, 4, and 8 threads for each hardware and additionally test 16 threads for Apple Silicons on Larq. And daBNN only supports single-thread inference. Results are showcased in Table 13-16.

Table 7: Accuracy on ShapeNet dataset.

| Task | Arch. | Category | FP32 | Binarization Algorithm | | | | | | | |
|------|-------|----------|------|------|------|--------|---------|--------|---------|------|------|
| | | | | BNN | XNOR | DoReFa | Bi-Real | XNOR++ | ReActNet | ReCU | FDA |
| | | Airplane | 83.7 | 37.5 | 74.14 | 67.2 | 67.61 | 30.36 | 66.12 | 31.61 | 65.34 |
| | | Bag | 79.6 | 44.2 | 49 | 55.34 | 47.11 | 37.44 | 50.28 | 38.58 | 48.62 |
| | | Cap | 92.3 | 44.3 | 73.32 | 51.21 | 61.41 | 40.37 | 56.73 | 40.13 | 56 |
| | | Car | 76.8 | 24.3 | 55.27 | 52.24 | 49.39 | 24.07 | 49.11 | 23.92 | 58.5 |
| | | Chair | 90.9 | 61.6 | 85.62 | 83.96 | 83.6 | 41.89 | 83.83 | 41.5 | 83.27 |
| | | Earphone | 70.2 | 38.5 | 30.97 | 34.94 | 35.24 | 26.3 | 36.72 | 23.01 | 34.46 |
| | | Guitar | 91.1 | 32.9 | 69.17 | 67.9 | 65.99 | 23.45 | 64.18 | 28.38 | 78.69 |
| | | Knife | 85.7 | 43 | 78.16 | 76.16 | 75.53 | 37.62 | 75.01 | 38.81 | 77.07 |
| ShapeNet | PointNet | Lamp | 82 | 51.2 | 69 | 68.75 | 60 | 49.45 | 66.13 | 48.41 | 67.45 |
| | | Laptop | 95.5 | 49.4 | 93.29 | 92.93 | 92.79 | 41.89 | 92.93 | 42.28 | 93.66 |
| | | Motorbike | 64.4 | 16.3 | 19.04 | 18.88 | 18.69 | 13.18 | 18.59 | 11.26 | 20.38 |
| | | Mug | 93.6 | 49.1 | 64.32 | 53.56 | 52.01 | 47.58 | 52.51 | 46.83 | 53.48 |
| | | Pistol | 80.8 | 25.5 | 62.29 | 59.15 | 51.43 | 26.96 | 53.79 | 27.81 | 62.61 |
| | | Rocket | 54.4 | 26.9 | 30.95 | 27.92 | 26.61 | 22.38 | 26.01 | 19.32 | 23.08 |
| | | Skateboard | 70.7 | 41.2 | 45.7 | 50.15 | 43.78 | 28.63 | 43.74 | 26.71 | 45.81 |
| | | Table | 81.4 | 51.3 | 73.68 | 72.69 | 69.72 | 45.74 | 69.56 | 45.21 | 73.45 |
| | | Overall | 80.81875 | 39.82 | 60.87 | 58.31 | 56.31 | 33.58 | 56.58 | 33.36 | 58.68 |

Table 8: Accuracy on Language and Speech Tasks.

| Task | Arch. | FP32 | Binarization Algorithm | | | | | | | |
|---|---|---|---|---|---|---|---|---|---|---|
| | | | BNN | XNOR | DoReFa | Bi-Real | XNOR++ | ReActNet | ReCU | FDA |
| | BERT-Tiny$_{4L}$ | 82.81 | 36.90 | 41.20 | 52.31 | 55.09 | 37.27 | 55.52 | 38.55 | 59.41 |
| MNLI-m | BERT-Tiny$_{6L}$ | 84.76 | 37.01 | 51.17 | 63.09 | 66.81 | 37.98 | 66.47 | 37.95 | 68.46 |
| | BERT-Base | 84.88 | 35.45 | 41.40 | 60.67 | 62.47 | 35.45 | 60.22 | 35.45 | 63.49 |
| | BERT-Tiny$_{4L}$ | 83.08 | 36.54 | 41.55 | 53.01 | 55.57 | 36.07 | 55.89 | 37.62 | 59.76 |
| MNLI-mm | BERT-Tiny$_{6L}$ | 84.42 | 36.47 | 50.92 | 63.87 | 66.82 | 38.11 | 67.64 | 36.91 | 68.98 |
| | BERT-Base | 85.45 | 35.22 | 41.18 | 60.96 | 63.17 | 35.22 | 61.19 | 35.22 | 63.72 |
| | BERT-Tiny$_{4L}$ | 90.47 | 66.19 | 73.69 | 75.79 | 77.38 | 64.97 | 76.92 | 67.32 | 78.92 |
| QQP | BERT-Tiny$_{6L}$ | 85.98 | 63.18 | 78.90 | 80.93 | 82.42 | 63.19 | 82.95 | 63.3 | 83.19 |
| | BERT-Base | 91.51 | 63.18 | 71.93 | 77.07 | 80.01 | 63.18 | 81.16 | 63.18 | 83.26 |
| | BERT-Tiny$_{4L}$ | 87.46 | 51.71 | 60.59 | 61.15 | 61.92 | 52.79 | 62.67 | 53.99 | 62.29 |
| QNLI | BERT-Tiny$_{6L}$ | 90.79 | 52.22 | 62.75 | 66.88 | 69.72 | 51.84 | 70.27 | 51.32 | 72.72 |
| | BERT-Base | 92.14 | 51.8 | 60.29 | 70.78 | 70.14 | 54.07 | 69.44 | 51.87 | 72.43 |
| GLUE | BERT-Tiny$_{4L}$ | 92.43 | 52.98 | 79.93 | 82.45 | 84.06 | 54.01 | 84.17 | 54.24 | 86.12 |
| SST-2 | BERT-Tiny$_{6L}$ | 90.25 | 58.14 | 84.74 | 86.23 | 87.73 | 69.38 | 87.95 | 52.40 | 87.72 |
| | BERT-Base | 93.23 | 52.29 | 78.78 | 86.01 | 86.35 | 53.32 | 84.4 | 52.40 | 87.93 |
| | BERT-Tiny$_{4L}$ | 49.61 | 6.55 | 7.22 | 12.69 | 16.86 | 0 | 14.71 | 6.25 | 17.80 |
| CoLA | BERT-Tiny$_{6L}$ | 54.17 | 2.57 | 12.57 | 15.97 | 17.94 | 0 | 15.24 | 2.24 | 22.21 |
| | BERT-Base | 59.71 | 4.63 | 0 | 4.74 | 15.95 | 0 | 4.63 | 0.40 | 4.63 |
| | BERT-Tiny$_{4L}$ | 86.35 | 4.31 | 18.05 | 18.74 | 22.65 | 7.45 | 22.73 | 8.20 | 27.56 |
| STS-B | BERT-Tiny$_{6L}$ | 89.79 | 1.04 | 14.72 | 22.31 | 24.59 | 5.70 | 23.40 | 8.22 | 37.15 |
| | BERT-Base | 90.06 | 6.94 | 12.19 | 18.26 | 20.76 | 4.99 | 8.73 | 6.59 | 10.14 |
| | BERT-Tiny$_{4L}$ | 85.50 | 68.30 | 71.74 | 71.99 | 71.74 | 68.30 | 71.74 | 71.25 | 71.49 |
| MRPC | BERT-Tiny$_{6L}$ | 87.71 | 68.30 | 70.76 | 71.74 | 71.49 | 68.30 | 71.74 | 69.04 | 71.74 |
| | BERT-Base | 86.24 | 68.30 | 68.3 | 70.02 | 70.27 | 68.30 | 71.25 | 68.30 | 69.04 |
| | BERT-Tiny$_{4L}$ | 65.34 | 56.31 | 53.43 | 56.31 | 55.59 | 54.15 | 57.76 | 61.01 | 59.20 |
| RTE | BERT-Tiny$_{6L}$ | 68.95 | 56.31 | 54.51 | 54.51 | 58.12 | 49.09 | 53.43 | 58.84 | 54.87 |
| | BERT-Base | 72.20 | 53.43 | 57.04 | 55.23 | 54.51 | 54.87 | 54.51 | 55.23 | 55.23 |
| Speech Commands | FSMN | 94.89 | 56.45 | 56.45 | 68.65 | 73.60 | 75.04 | 73.80 | 56.45 | 56.45 |
| | D-FSMN | 97.51 | 88.32 | 92.03 | 78.92 | 85.11 | 56.77 | 83.80 | 92.11 | 93.91 |

Table 9: Results for Robustness Corruption on CIFAR10-C Dataset with Different Binarization Algorithms (1/2).

| Noise | FP32 | Binarization Algorithm | | | | | | | |
|---|---|---|---|---|---|---|---|---|---|
| | | BNN | XNOR | DoReFa | Bi-Real | XNOR++ | ReActNet | ReCU | FDA |
| Origin | 94.82 | 89.69 | 91.40 | 91.55 | 91.20 | 90.04 | 91.55 | 92.79 | 90.42 |
| gaussian_noise-1 | 78.23 | 74.22 | 76.00 | 74.97 | 74.95 | 75.07 | 75.15 | 78.25 | 77.36 |
| gaussian_noise-2 | 56.72 | 56.73 | 62.44 | 55.94 | 58.33 | 57.52 | 55.97 | 61.32 | 60.48 |
| gaussian_noise-3 | 36.93 | 42.69 | 47.58 | 39.56 | 43.47 | 40.79 | 37.99 | 43.32 | 44.26 |
| gaussian_noise-4 | 31.03 | 38.35 | 41.43 | 33.24 | 36.65 | 34.68 | 31.47 | 35.91 | 37.30 |
| gaussian_noise-5 | 25.54 | 34.05 | 36.22 | 28.66 | 31.78 | 30.13 | 25.49 | 30.19 | 32.09 |
| ipulse_nosie-1 | 82.54 | 84.57 | 86.94 | 84.68 | 87.30 | 85.72 | 86.89 | 88.73 | 85.8 |
| ipulse_nosie-2 | 70.12 | 77.13 | 80.74 | 77.35 | 81.14 | 79.62 | 80.25 | 82.80 | 80.3 |
| ipulse_nosie-3 | 59.88 | 70.58 | 75.01 | 69.20 | 74.59 | 71.82 | 72.16 | 76.05 | 72.6 |
| ipulse_nosie-4 | 40.59 | 54.42 | 59.39 | 49.48 | 56.66 | 52.61 | 49.79 | 58.44 | 56.45 |
| ipulse_nosie-5 | 26.03 | 39.86 | 41.54 | 32.72 | 37.28 | 35.12 | 28.42 | 38.26 | 39.98 |
| shot_noise-1 | 85.75 | 81.51 | 81.31 | 81.84 | 81.58 | 80.66 | 81.88 | 83.98 | 82.42 |
| shot_noise-2 | 76.61 | 72.04 | 74.02 | 72.21 | 72.81 | 73.03 | 72.32 | 76.70 | 75.1 |
| shot_noise-3 | 52.21 | 53.90 | 57.08 | 50.66 | 54.59 | 53.76 | 51.31 | 57.22 | 56.56 |
| shot_noise-4 | 44.13 | 47.58 | 51.29 | 43.59 | 48.36 | 46.64 | 44.21 | 48.78 | 48.91 |
| shot_noise-5 | 32.73 | 39.93 | 40.79 | 33.80 | 38.50 | 36.47 | 31.79 | 36.46 | 37.8 |
| speckle_noise-1 | 86.30 | 81.29 | 81.94 | 80.93 | 80.77 | 81.14 | 82.17 | 84.17 | 82.62 |
| speckle_noise-2 | 71.94 | 68.07 | 70.14 | 67.5 | 69.22 | 69.35 | 68.26 | 72.94 | 71.70 |
| speckle_noise-3 | 64.47 | 62.12 | 64.13 | 60.24 | 63.44 | 62.50 | 61.14 | 66.89 | 64.27 |
| speckle_noise-4 | 49.81 | 51.93 | 53.77 | 47.93 | 52.75 | 50.59 | 48.39 | 54.13 | 52.40 |
| speckle_noise-5 | 38.70 | 44.25 | 43.60 | 38.65 | 43.16 | 42.09 | 37.78 | 42.13 | 42.57 |
| gaussian_blur-1 | 94.17 | 89.03 | 90.5 | 89.33 | 90.56 | 89.00 | 91.05 | 92.16 | 89.33 |
| gaussian_blur-2 | 87.04 | 78.3 | 81.98 | 78.81 | 80.42 | 77.75 | 81.20 | 84.80 | 78.93 |
| gaussian_blur-3 | 75.15 | 67.74 | 68.27 | 67.67 | 68.16 | 64.54 | 67.42 | 73.62 | 66.29 |
| gaussian_blur-4 | 59.5 | 55.17 | 53.63 | 55.74 | 54.08 | 52.44 | 52.72 | 60.32 | 53.37 |
| gaussian_blur-5 | 36.03 | 37.31 | 33.96 | 37.50 | 37.54 | 36.77 | 34.08 | 39.22 | 34.93 |
| defocus_blur-1 | 94.2 | 88.73 | 91.06 | 89.1 | 90.32 | 88.91 | 90.92 | 91.98 | 89.58 |
| defocus_blur-2 | 92.75 | 85.97 | 88.99 | 86.59 | 88.31 | 85.58 | 87.91 | 90.47 | 87.01 |
| defocus_blur-3 | 87.38 | 79.02 | 82.43 | 78.88 | 80.71 | 77.58 | 80.88 | 84.85 | 79.52 |
| defocus_blur-4 | 76.99 | 69.13 | 71.02 | 68.29 | 68.33 | 65.96 | 68.42 | 74.40 | 68.22 |
| defocus_blur-5 | 52.09 | 48.85 | 51.99 | 48.82 | 49.17 | 48.45 | 46.92 | 55.70 | 48.27 |
| glass_blur-1 | 54.93 | 56.57 | 51.72 | 57.94 | 56.78 | 57.29 | 56.27 | 58.82 | 58.92 |
| glass_blur-2 | 56.37 | 57.93 | 53.46 | 60.42 | 59.21 | 59.32 | 58.03 | 60.25 | 60.56 |
| glass_blur-3 | 59.21 | 61.43 | 56.98 | 64.11 | 61.72 | 62.41 | 60.39 | 62.84 | 63.32 |
| glass_blur-4 | 45.65 | 46.50 | 42.72 | 48.48 | 47.19 | 47.83 | 46.88 | 49.09 | 49.23 |
| glass_blur-5 | 49.19 | 49.52 | 46.40 | 52.06 | 49.83 | 50.02 | 49.08 | 51.14 | 51.82 |
| otion_blur-1 | 89.40 | 81.57 | 83.27 | 82.00 | 83.11 | 81.48 | 84.19 | 86.21 | 82.83 |
| otion_blur-2 | 81.95 | 71.52 | 74.71 | 73.38 | 72.48 | 70.99 | 74.35 | 77.75 | 74.09 |
| otion_blur-3 | 72.48 | 61.87 | 66.21 | 63.86 | 63.39 | 61.57 | 63.85 | 68.31 | 64.36 |
| otion_blur-4 | 72.79 | 62.40 | 66.18 | 63.94 | 62.84 | 62.03 | 64.54 | 67.88 | 64.13 |
| otion_blur-5 | 63.91 | 54.14 | 57.98 | 56.07 | 55.90 | 54.35 | 55.60 | 59.71 | 56.65 |
| zoo_blur-1 | 87.36 | 78.25 | 81.31 | 78.56 | 79.45 | 77.20 | 80.22 | 83.69 | 78.55 |
| zoo_blur-2 | 83.89 | 74.84 | 77.73 | 75.39 | 75.88 | 72.83 | 75.74 | 80.46 | 74.72 |
| zoo_blur-3 | 77.73 | 69.00 | 70.98 | 68.81 | 69.03 | 66.56 | 68.34 | 74.33 | 68.21 |
| zoo_blur-4 | 71.39 | 64.12 | 65.21 | 63.79 | 62.81 | 61.01 | 62.47 | 68.67 | 62.58 |
| zoo_blur-5 | 60.60 | 55.15 | 55.83 | 55.4 | 54.38 | 52.66 | 52.24 | 59.51 | 53.94 |
| brighness-1 | 94.31 | 89.29 | 90.84 | 89.53 | 90.8 | 89.30 | 90.97 | 92.06 | 89.74 |
| brighness-2 | 94.03 | 88.25 | 90.42 | 88.71 | 89.66 | 88.50 | 90.64 | 91.64 | 88.77 |
| brighness-3 | 93.53 | 87.40 | 89.38 | 87.38 | 89.17 | 87.31 | 89.63 | 90.72 | 87.84 |
| brighness-4 | 92.74 | 85.45 | 88.27 | 86.12 | 87.78 | 85.44 | 88.16 | 89.58 | 86.39 |
| brighness-5 | 90.36 | 80.95 | 85.22 | 81.65 | 83.79 | 80.99 | 84.45 | 86.53 | 82.04 |

Table 10: Results for Robustness Corruption on CIFAR10-C Dataset with Different Binarization Algorithms (2/2).

| Noise | FP32 | Binarization Algorithm | | | | | | | |
| | | BNN | XNOR | DoReFa | Bi-Real | XNOR++ | ReActNet | ReCU | FDA |
|---|---|---|---|---|---|---|---|---|---|
| fog-1 | 94.04 | 88.17 | 90.89 | 88.84 | 89.91 | 88.51 | 90.84 | 92.08 | 89.43 |
| fog-2 | 93.03 | 84.58 | 88.85 | 85.48 | 87.26 | 84.77 | 88 | 89.87 | 86.76 |
| fog-3 | 90.69 | 78.07 | 85.2 | 80.07 | 83.32 | 78.94 | 83.77 | 86.82 | 82.78 |
| fog-4 | 86.72 | 69.56 | 78.92 | 72.27 | 75.89 | 71.01 | 77.96 | 81.56 | 75.62 |
| fog-5 | 68.6 | 49.04 | 53.9 | 52.33 | 52.88 | 49.68 | 57.67 | 62.29 | 55.18 |
| frost-1 | 89.97 | 83.66 | 84.7 | 84.07 | 85.76 | 83.64 | 85.51 | 87.75 | 84.85 |
| frost-2 | 84.42 | 77.88 | 78.97 | 77.47 | 78.96 | 77.26 | 79.14 | 81.4 | 79.16 |
| frost-3 | 74.85 | 67.67 | 69.3 | 67.14 | 68.76 | 66.03 | 69.58 | 72.54 | 70.14 |
| frost-4 | 73.32 | 65.93 | 67.14 | 65.97 | 68.52 | 65.37 | 67.93 | 71.44 | 69.41 |
| frost-5 | 62.13 | 55.02 | 56.67 | 55.62 | 56.77 | 54.26 | 57.69 | 61.11 | 59.88 |
| snow-1 | 89.26 | 84.58 | 85.44 | 84.59 | 86.43 | 85.07 | 85.95 | 87.82 | 85.67 |
| snow-2 | 78.96 | 72.01 | 73.37 | 73.14 | 73.42 | 72.43 | 73.65 | 78.05 | 73.84 |
| snow-3 | 82.85 | 75.6 | 76.84 | 75.74 | 76.95 | 75.74 | 77.76 | 79.98 | 75.95 |
| snow-4 | 80.29 | 70.84 | 72.56 | 71.93 | 72.39 | 71.25 | 72.97 | 76.12 | 72.16 |
| snow-5 | 74.94 | 63.85 | 66.94 | 65.9 | 65.71 | 64.24 | 66.29 | 70.33 | 66.53 |
| contrast-1 | 93.82 | 87.23 | 89.96 | 87.93 | 89.68 | 87.88 | 90.16 | 91.53 | 88.84 |
| contrast-2 | 90.53 | 76.02 | 84 | 77.27 | 82.1 | 76.3 | 82.8 | 86.02 | 81.37 |
| contrast-3 | 85.84 | 63.97 | 77.1 | 65.77 | 72.77 | 64.18 | 74.62 | 79.30 | 71.89 |
| contrast-4 | 75.08 | 44.07 | 62.37 | 47.35 | 55.57 | 44.94 | 59.87 | 65.72 | 55.14 |
| contrast-5 | 29.36 | 20 | 25.67 | 20.18 | 22.18 | 21.04 | 25.28 | 25.66 | 24.04 |
| elastic_transfor-1 | 89.97 | 82.97 | 84.5 | 83.38 | 84.74 | 82.48 | 84.42 | 86.54 | 83.25 |
| elastic_transfor-2 | 89.43 | 82.12 | 84.79 | 82.44 | 84.07 | 81.97 | 84.61 | 86.20 | 83.15 |
| elastic_transfor-3 | 85.52 | 77.56 | 80.71 | 77.92 | 79.11 | 77 | 79.53 | 82.27 | 78.35 |
| elastic_transfor-4 | 79.48 | 73.75 | 74.83 | 73.92 | 73.41 | 72.77 | 73.98 | 77.53 | 74.17 |
| elastic_transfor-5 | 75.02 | 70.97 | 70.22 | 71.36 | 71.03 | 71.63 | 71.65 | 75.31 | 71.49 |
| jpeg_copression-1 | 87.36 | 83.28 | 83.93 | 84.07 | 83.83 | 83.77 | 84.3 | 85.65 | 83.63 |
| jpeg_copression-2 | 81.68 | 80.09 | 79.66 | 79.77 | 80.03 | 80.29 | 80.59 | 81.66 | 79.8 |
| jpeg_copression-3 | 79.98 | 78.55 | 78.21 | 78.32 | 78.27 | 78.99 | 78.51 | 79.94 | 78.44 |
| jpeg_copression-4 | 77.17 | 77.12 | 75.78 | 77.44 | 77.04 | 77.46 | 77.08 | 77.67 | 76.71 |
| jpeg_copression-5 | 73.85 | 74.51 | 73.04 | 74.65 | 74.13 | 75.26 | 74.16 | 74.85 | 74.19 |
| pixelate-1 | 92.57 | 86.97 | 88.19 | 87.39 | 88.73 | 87.47 | 88.17 | 89.42 | 87.26 |
| pixelate-2 | 88.23 | 81.91 | 80.95 | 82.37 | 82.82 | 81.93 | 81.99 | 83.80 | 80.95 |
| pixelate-3 | 84 | 78.4 | 75.25 | 78.63 | 78.03 | 77.15 | 77.28 | 78.89 | 75.30 |
| pixelate-4 | 68.51 | 64.11 | 58.11 | 62.49 | 60.89 | 61.43 | 60.06 | 61.71 | 59.95 |
| pixelate-5 | 50.57 | 50.68 | 44.77 | 48.18 | 45.44 | 46.74 | 43.27 | 47.29 | 45.62 |
| saturate-1 | 92.41 | 84.98 | 88.38 | 85.38 | 87.26 | 85.39 | 88.23 | 89.34 | 86.82 |
| saturate-2 | 90.12 | 80.74 | 85.26 | 81.57 | 82.68 | 80.74 | 84.22 | 86.06 | 83.09 |
| saturate-3 | 93.83 | 87.89 | 90.45 | 88.12 | 89.53 | 88.03 | 90.15 | 90.97 | 88.73 |
| saturate-4 | 91.61 | 82.5 | 86.88 | 82.6 | 84.66 | 82.44 | 86.04 | 87.56 | 83.70 |
| saturate-5 | 87.48 | 76.03 | 82.76 | 75.53 | 78.3 | 75.85 | 80.62 | 82.64 | 77.00 |
| spatter-1 | 91.17 | 87.5 | 89.75 | 87.83 | 89.34 | 87.9 | 89.42 | 91.00 | 88.98 |
| spatter-2 | 85.2 | 83.85 | 85.98 | 83.52 | 85.64 | 84.72 | 85.88 | 87.59 | 85.00 |
| spatter-3 | 80.63 | 77.94 | 80.33 | 77.95 | 80.19 | 79.41 | 80.07 | 82.65 | 79.50 |
| spatter-4 | 94.68 | 84.57 | 86.71 | 84.77 | 86.51 | 85.14 | 86.72 | 88.22 | 85.32 |
| spatter-5 | 74.07 | 78.85 | 80.77 | 78.71 | 80.94 | 79.71 | 80.51 | 83.14 | 79.48 |
| Overall | 74.11 | 69.43 | 71.51 | 69.36 | 70.76 | 69.09 | 70.31 | 73.56 | 70.70 |

Table 11: Sensitivity to Hyper Parameters in Training (1/2).

| Algorithm | Epoch | Optimizer | Learning Rate | Scheduler | Acc. | $\text{Acc.}_1$ | $\text{Acc.}_2$ | $\text{Acc.}_3$ | $\text{Acc.}_4$ | mean | std |
|---|---|---|---|---|---|---|---|---|---|---|---|
| FP32 | 200 | SGD | 0.1 | cosine | 94.58 | 94.6 | 95.05 | 94.64 | 94.84 | 94.74 | 0.20 |
| | 200 | SGD | 0.1 | step | 92.63 | 92.42 | 92.15 | 92.62 | 92.38 | 92.44 | 0.20 |
| | 200 | SGD | 0.01 | cosine | 92.23 | 91.76 | 91.76 | 91.99 | 92.17 | 91.98 | 0.22 |
| | 200 | SGD | 0.01 | step | 83.94 | 83.50 | 82.80 | 84.13 | 83.89 | 83.65 | 0.53 |
| | 200 | Adam | 0.001 | cosine | 93.51 | 92.94 | 93.12 | 93.35 | 92.86 | 93.16 | 0.27 |
| | 200 | Adam | 0.001 | step | 93.37 | 93.15 | 93.32 | 93.41 | 93.35 | 93.32 | 0.10 |
| | 200 | Adam | 0.0001 | cosine | 89.97 | 89.92 | 89.96 | 89.9 | 89.92 | 89.93 | 0.03 |
| | 200 | Adam | 0.0001 | step | 90.57 | 89.91 | 90.43 | 90.25 | 90.31 | 90.29 | 0.25 |
| BNN | 200 | SGD | 0.1 | cosine | 87.62 | 87.53 | 87.99 | 88.86 | 87.84 | 87.97 | 0.53 |
| | 200 | SGD | 0.1 | step | 70.87 | 73.86 | 71.83 | 73.1 | 72.87 | 72.51 | 1.17 |
| | 200 | SGD | 0.01 | cosine | 73.52 | 72.62 | 72.82 | 71.14 | 72.59 | 72.54 | 0.87 |
| | 200 | SGD | 0.01 | step | 52.85 | 51.77 | 52.00 | 52.34 | 53.14 | 52.42 | 0.57 |
| | 200 | Adam | 0.001 | cosine | 88.76 | 88.99 | 88.67 | 88.84 | 88.81 | 88.81 | 0.12 |
| | 200 | Adam | 0.001 | step | 88.85 | 89.34 | 88.77 | 89.02 | 89.00 | 89.00 | 0.22 |
| | 200 | Adam | 0.0001 | cosine | 83.46 | 83.09 | 83.20 | 83.70 | 83.20 | 83.33 | 0.25 |
| | 200 | Adam | 0.0001 | step | 84.08 | 84.11 | 84.20 | 84.31 | 83.56 | 84.05 | 0.29 |
| XNOR | 200 | SGD | 0.1 | cosine | 91.83 | 91.99 | 91.87 | 92.01 | 91.56 | 91.85 | 0.18 |
| | 200 | SGD | 0.1 | step | 90.02 | 90.01 | 90.12 | 89.82 | 89.7 | 89.93 | 0.17 |
| | 200 | SGD | 0.01 | cosine | 90.09 | 89.68 | 90.01 | 89.97 | 90.00 | 89.95 | 0.16 |
| | 200 | SGD | 0.01 | step | 86.86 | 86.66 | 87.21 | 86.98 | 86.61 | 86.86 | 0.24 |
| | 200 | Adam | 0.001 | cosine | 89.39 | 89.81 | 89.73 | 89.91 | 89.75 | 89.72 | 0.20 |
| | 200 | Adam | 0.001 | step | 89.92 | 89.79 | 89.73 | 90.01 | 89.61 | 89.81 | 0.16 |
| | 200 | Adam | 0.0001 | cosine | 86.18 | 86.29 | 87.03 | 86.36 | 86.62 | 86.50 | 0.34 |
| | 200 | Adam | 0.0001 | step | 86.32 | 87.04 | 86.68 | 86.99 | 87.18 | 86.84 | 0.34 |
| DoReFa | 200 | SGD | 0.1 | cosine | 85.64 | 85.67 | 85.89 | 86.00 | 85.79 | 85.80 | 0.15 |
| | 200 | SGD | 0.1 | step | 86.95 | 86.98 | 86.69 | 86.62 | 86.65 | 86.78 | 0.17 |
| | 200 | SGD | 0.01 | cosine | 86.56 | 86.59 | 86.52 | 86.69 | 86.88 | 86.65 | 0.14 |
| | 200 | SGD | 0.01 | step | 78.76 | 79.97 | 80.73 | 79.94 | 80.47 | 79.97 | 0.76 |
| | 200 | Adam | 0.001 | cosine | 88.85 | 89.06 | 88.92 | 88.87 | 88.75 | 88.89 | 0.11 |
| | 200 | Adam | 0.001 | step | 89.08 | 89.16 | 88.93 | 89.23 | 88.84 | 89.05 | 0.16 |
| | 200 | Adam | 0.0001 | cosine | 83.56 | 83.17 | 83.65 | 83.60 | 83.66 | 83.53 | 0.20 |
| | 200 | Adam | 0.0001 | step | 83.70 | 83.74 | 84.27 | 84.19 | 84.01 | 83.98 | 0.26 |
| Bi-Real | 200 | SGD | 0.1 | cosine | 87.55 | 87.81 | 88.06 | 87.30 | 87.88 | 87.72 | 0.30 |
| | 200 | SGD | 0.1 | step | 87.95 | 88.35 | 88.13 | 87.73 | 88.25 | 88.08 | 0.25 |
| | 200 | SGD | 0.01 | cosine | 87.76 | 87.93 | 87.73 | 87.72 | 87.64 | 87.76 | 0.11 |
| | 200 | SGD | 0.01 | step | 83.75 | 82.91 | 82.82 | 82.91 | 83.39 | 83.16 | 0.40 |
| | 200 | Adam | 0.001 | cosine | 88.78 | 89.15 | 89.06 | 89.00 | 89.2 | 89.04 | 0.16 |
| | 200 | Adam | 0.001 | step | 88.89 | 88.98 | 88.78 | 89.11 | 89.05 | 88.96 | 0.13 |
| | 200 | Adam | 0.0001 | cosine | 83.96 | 84.17 | 84.37 | 83.54 | 84.07 | 84.02 | 0.31 |
| | 200 | Adam | 0.0001 | step | 84.63 | 84.48 | 84.32 | 84.75 | 84.29 | 84.49 | 0.20 |

Table 12: Sensitivity to Hyper Parameters in Training (2/2).

| Algorithm | Epoch | Optimizer | Learning Rate | Scheduler | Acc. | Acc._1 | Acc._2 | Acc._3 | Acc._4 | mean | std |
|---|---|---|---|---|---|---|---|---|---|---|---|
| XNOR++ | 200 | SGD | 0.1 | cosine | 87.82 | 88.42 | 88.12 | 88.55 | 88.19 | 88.22 | 0.28 |
| | 200 | SGD | 0.1 | step | 73.55 | 73.11 | 75.06 | 74.05 | 73.78 | 73.91 | 0.73 |
| | 200 | SGD | 0.01 | cosine | 74.03 | 75.06 | 73.64 | 74.53 | 74.71 | 74.39 | 0.56 |
| | 200 | SGD | 0.01 | step | 53.55 | 54.16 | 54.01 | 52.91 | 54.36 | 53.80 | 0.58 |
| | 200 | Adam | 0.001 | cosine | 88.77 | 88.65 | 89.10 | 88.61 | 88.81 | 88.79 | 0.19 |
| | 200 | Adam | 0.001 | step | 89.18 | 89.05 | 89.27 | 88.93 | 89.00 | 89.09 | 0.14 |
| | 200 | Adam | 0.0001 | cosine | 83.86 | 83.49 | 83.56 | 83.16 | 83.62 | 83.54 | 0.25 |
| | 200 | Adam | 0.0001 | step | 83.46 | 83.77 | 84.40 | 84.06 | 83.82 | 83.90 | 0.35 |
| ReActNet | 200 | SGD | 0.1 | cosine | 88.60 | 88.53 | 88.38 | 88.48 | 88.89 | 88.58 | 0.19 |
| | 200 | SGD | 0.1 | step | 88.42 | 88.01 | 88.10 | 88.02 | 88.43 | 88.20 | 0.21 |
| | 200 | SGD | 0.01 | cosine | 87.75 | 87.86 | 88.00 | 87.80 | 88.02 | 87.89 | 0.12 |
| | 200 | SGD | 0.01 | step | 83.29 | 82.89 | 83.65 | 83.76 | 83.27 | 83.37 | 0.35 |
| | 200 | Adam | 0.001 | cosine | 89.47 | 89.29 | 89.01 | 89.05 | 89.14 | 89.19 | 0.19 |
| | 200 | Adam | 0.001 | step | 89.27 | 89.74 | 89.48 | 89.40 | 89.39 | 89.46 | 0.18 |
| | 200 | Adam | 0.0001 | cosine | 84.65 | 84.93 | 84.48 | 84.65 | 84.67 | 84.68 | 0.16 |
| | 200 | Adam | 0.0001 | step | 84.69 | 84.55 | 84.93 | 84.94 | 85.38 | 84.90 | 0.32 |
| ReCU | 200 | SGD | 0.1 | cosine | 91.72 | 91.94 | 91.68 | 91.69 | 91.81 | 91.77 | 0.11 |
| | 200 | SGD | 0.1 | step | 87.73 | 88.14 | 87.81 | 88.02 | 87.91 | 87.92 | 0.16 |
| | 200 | SGD | 0.01 | cosine | 87.32 | 87.28 | 87.53 | 87.48 | 87.32 | 87.39 | 0.11 |
| | 200 | SGD | 0.01 | step | 71.86 | 71.72 | 71.78 | 72.26 | 71.59 | 71.84 | 0.25 |
| | 200 | Adam | 0.001 | cosine | 88.24 | 89.98 | 88.26 | 88.48 | 88.13 | 88.62 | 0.77 |
| | 200 | Adam | 0.001 | step | 88.36 | 88.48 | 88.55 | 88.42 | 88.63 | 88.49 | 0.11 |
| | 200 | Adam | 0.0001 | cosine | 80.07 | 81.10 | 80.62 | 81.09 | 79.95 | 80.57 | 0.55 |
| | 200 | Adam | 0.0001 | step | 81.26 | 81.42 | 81.08 | 81.58 | 81.69 | 81.41 | 0.24 |
| FDA | 200 | SGD | 0.1 | cosine | 89.69 | 89.59 | 89.56 | 89.53 | 89.65 | 89.60 | 0.07 |
| | 200 | SGD | 0.1 | step | 80.38 | 80.34 | 80.83 | 80.52 | 80.52 | 80.52 | 0.19 |
| | 200 | SGD | 0.01 | cosine | 80.72 | 80.93 | 80.89 | 80.70 | 80.79 | 80.81 | 0.10 |
| | 200 | SGD | 0.01 | step | 63.41 | 62.85 | 63.04 | 63.04 | 63.14 | 63.10 | 0.20 |
| | 200 | Adam | 0.001 | cosine | 89.70 | 89.57 | 89.57 | 89.80 | 89.76 | 89.68 | 0.11 |
| | 200 | Adam | 0.001 | step | 89.84 | 89.85 | 90.10 | 89.79 | 90.01 | 89.92 | 0.13 |
| | 200 | Adam | 0.0001 | cosine | 89.59 | 89.10 | 89.34 | 89.31 | 89.51 | 89.37 | 0.19 |
| | 200 | Adam | 0.0001 | step | 89.52 | 89.59 | 89.52 | 89.64 | 89.58 | 89.57 | 0.05 |

Table 13: Inference Efficiency on Hardware (1/4).

| Hardware | Threads | Arch. | Larq | | | daBNN | |
|---|---|---|---|---|---|---|---|
| | | | FP32 | BNN | ReAct | FP32 | BNN |
| Kirin 970 | 1 | ResNet18 | 716.427 | 123.263 | 126.457 | 427.585 | 72.585 |
| | | ResNet34 | 1449.67 | 159.615 | 171.227 | 836.321 | 124.091 |
| | | VGG-Small | 242.443 | 14.833 | 16.401 | – | – |
| | 2 | ResNet18 | 372.642 | 72.697 | 78.605 | – | – |
| | | ResNet34 | 732.355 | 96.711 | 108.41 | – | – |
| | | VGG-Small | 121.91 | 10.304 | 11.935 | – | – |
| | 4 | ResNet18 | 191.517 | 42.986 | 47.182 | – | – |
| | | ResNet34 | 367.891 | 61.413 | 73.101 | – | – |
| | | VGG-Small | 57.721 | 8.72 | 8.387 | – | – |
| | 8 | ResNet18 | 96.937 | 37.457 | 56.017 | – | – |
| | | ResNet34 | 212.982 | 53.809 | 67.667 | – | – |
| | | VGG-Small | 33.647 | 18.649 | 19.818 | – | – |
| Kirin 980 | 1 | ResNet18 | 307.624 | 49.009 | 50.018 | 158.363 | 31.803 |
| | | ResNet34 | 507.734 | 71.909 | 74.920 | 308.537 | 53.031 |
| | | VGG-Small | 83.163 | 7.772 | 8.215 | – | – |
| | 2 | ResNet18 | 187.494 | 52.057 | 54.285 | – | – |
| | | ResNet34 | 367.853 | 57.336 | 60.483 | – | – |
| | | VGG-Small | 49.264 | 6.116 | 5.604 | – | – |
| | 4 | ResNet18 | 104.076 | 29.556 | 35.539 | – | – |
| | | ResNet34 | 202.173 | 31.324 | 35.911 | – | – |
| | | VGG-Small | 22.690 | 3.147 | 3.291 | – | – |
| | 8 | ResNet18 | 60.307 | 45.683 | 56.416 | – | – |
| | | ResNet34 | 120.738 | 60.758 | 86.887 | – | – |
| | | VGG-Small | 18.147 | 21.688 | 23.350 | – | – |
| Kirin 985 | 1 | ResNet18 | 173.238 | 27.429 | 30.626 | 164.556 | 34.528 |
| | | ResNet34 | 438.971 | 58.165 | 60.885 | 323.439 | 57.808 |
| | | VGG-Small | 70.797 | 6.147 | 6.796 | – | – |
| | 2 | ResNet18 | 103.621 | 25.672 | 35.477 | – | – |
| | | ResNet34 | 327.416 | 53.949 | 62.865 | – | – |
| | | VGG-Small | 55.328 | 5.955 | 6.243 | – | – |
| | 4 | ResNet18 | 92.387 | 26.728 | 34.778 | – | – |
| | | ResNet34 | 184.050 | 39.881 | 52.153 | – | – |
| | | VGG-Small | 28.076 | 8.919 | 14.795 | – | – |
| | 8 | ResNet18 | 130.972 | 82.772 | 89.766 | – | – |
| | | ResNet34 | 227.504 | 119.586 | 143.958 | – | – |
| | | VGG-Small | 44.339 | 34.034 | 43.816 | – | – |
| Kirin 990 | 1 | ResNet18 | 114.238 | 21.235 | 22.066 | 144.205 | 29.239 |
| | | ResNet34 | 227.043 | 31.545 | 32.821 | 275.502 | 49.476 |
| | | VGG-Small | 38.118 | 3.338 | 3.482 | – | – |
| | 2 | ResNet18 | 59.329 | 13.911 | 14.179 | – | – |
| | | ResNet34 | 116.822 | 23.452 | 22.770 | – | – |
| | | VGG-Small | 20.055 | 2.080 | 2.194 | – | – |
| | 4 | ResNet18 | 38.403 | 10.280 | 12.208 | – | – |
| | | ResNet34 | 81.273 | 15.570 | 17.727 | – | – |
| | | VGG-Small | 13.508 | 1.542 | 1.760 | – | – |
| | 8 | ResNet18 | 37.703 | 25.360 | 31.365 | – | – |
| | | ResNet34 | 78.753 | 34.884 | 39.363 | – | – |
| | | VGG-Small | 12.707 | 14.414 | 21.749 | – | – |

Table 14: Inference Efficiency on Hardware (2/4).

| Hardware | Threads | Arch. | Larq | | | daBNN | |
|---|---|---|---|---|---|---|---|
| | | | FP32 | BNN | ReAct | FP32 | BNN |
| Kirin 9000E | 1 | ResNet18 | 118.059 | 19.865 | 20.547 | 129.270 | 24.781 |
| | | ResNet34 | 236.047 | 31.822 | 32.575 | 250.680 | 42.134 |
| | | VGG-Small | 39.114 | 3.595 | 3.832 | – | – |
| | 2 | ResNet18 | 68.351 | 16.821 | 16.115 | – | – |
| | | ResNet34 | 133.671 | 25.061 | 24.660 | – | – |
| | | VGG-Small | 23.018 | 2.684 | 2.598 | – | – |
| | 4 | ResNet18 | 45.592 | 17.452 | 18.847 | – | – |
| | | ResNet34 | 91.648 | 23.395 | 28.022 | – | – |
| | | VGG-Small | 14.360 | 2.762 | 2.782 | – | – |
| | 8 | ResNet18 | 43.363 | 61.263 | 42.328 | – | – |
| | | ResNet34 | 89.405 | 70.232 | 93.558 | – | – |
| | | VGG-Small | 19.070 | 17.351 | 23.825 | – | – |
| Dimensity 820 | 1 | ResNet18 | 158.835 | 32.636 | 34.912 | 323.035 | 63.471 |
| | | ResNet34 | 328.020 | 57.133 | 60.807 | 629.493 | 102.443 |
| | | VGG-Small | 82.417 | 5.958 | 6.420 | – | – |
| | 2 | ResNet18 | 122.167 | 29.306 | 34.384 | – | – |
| | | ResNet34 | 250.088 | 43.306 | 50.143 | – | – |
| | | VGG-Small | 51.320 | 4.670 | 5.053 | – | – |
| | 4 | ResNet18 | 94.636 | 21.850 | 30.027 | – | – |
| | | ResNet34 | 177.757 | 33.809 | 40.816 | – | – |
| | | VGG-Small | 45.056 | 4.223 | 4.546 | – | – |
| | 8 | ResNet18 | 90.210 | 45.357 | 61.981 | – | – |
| | | ResNet34 | 166.989 | 68.444 | 74.286 | – | – |
| | | VGG-Small | 32.971 | 21.344 | 23.706 | – | – |
| Dimensity 9000 | 1 | ResNet18 | 106.388 | 21.023 | 24.770 | 148.690 | 29.030 |
| | | ResNet34 | 210.665 | 32.841 | 34.590 | 284.438 | 49.854 |
| | | VGG-Small | 42.057 | 4.410 | 5.530 | – | – |
| | 2 | ResNet18 | 81.606 | 22.661 | 27.050 | – | – |
| | | ResNet34 | 143.349 | 27.666 | 37.910 | – | – |
| | | VGG-Small | 26.512 | 2.273 | 2.410 | – | – |
| | 4 | ResNet18 | 51.421 | 13.079 | 15.200 | – | – |
| | | ResNet34 | 100.249 | 23.314 | 25.920 | – | – |
| | | VGG-Small | 17.735 | 3.015 | 3.770 | – | – |
| | 8 | ResNet18 | 43.355 | 24.939 | 30.740 | – | – |
| | | ResNet34 | 84.182 | 30.212 | 39.990 | – | – |
| | | VGG-Small | 14.857 | 14.258 | 17.540 | – | – |
| Snapdragon 855+ | 1 | ResNet18 | 90.430 | 19.769 | 20.530 | 163.293 | 31.174 |
| | | ResNet34 | 186.694 | 29.126 | 30.512 | 298.882 | 49.948 |
| | | VGG-Small | 29.735 | 3.153 | 3.259 | – | – |
| | 2 | ResNet18 | 58.510 | 25.780 | 26.331 | – | – |
| | | ResNet34 | 124.580 | 31.023 | 32.646 | – | – |
| | | VGG-Small | 19.408 | 2.258 | 2.471 | – | – |
| | 4 | ResNet18 | 39.269 | 19.865 | 23.297 | – | – |
| | | ResNet34 | 82.180 | 30.248 | 31.387 | – | – |
| | | VGG-Small | 13.566 | 2.032 | 2.359 | – | – |
| | 8 | ResNet18 | 36.630 | 49.060 | 85.861 | – | – |
| | | ResNet34 | 73.513 | 41.131 | 88.101 | – | – |
| | | VGG-Small | 12.860 | 17.828 | 23.489 | – | – |

Table 15: Inference Efficiency on Hardware (3/4).

| Hardware | Threads | Arch. | Larq | | | daBNN | |
| | | | FP32 | BNN | ReAct | FP32 | BNN |
|---|---|---|---|---|---|---|---|
| Snapdragon 870 | 1 | ResNet18 | 88.145 | 16.527 | 17.020 | 126.762 | 25.240 |
| | | ResNet34 | 185.468 | 25.488 | 26.195 | 237.361 | 41.440 |
| | | VGG-Small | 30.318 | 2.851 | 2.964 | – | – |
| | 2 | ResNet18 | 63.829 | 18.351 | 19.575 | – | – |
| | | ResNet34 | 159.174 | 25.352 | 26.340 | – | – |
| | | VGG-Small | 27.669 | 2.094 | 2.308 | – | – |
| | 4 | ResNet18 | 42.796 | 17.578 | 21.083 | – | – |
| | | ResNet34 | 89.960 | 25.816 | 27.201 | – | – |
| | | VGG-Small | 14.829 | 2.614 | 2.215 | – | – |
| | 8 | ResNet18 | 46.798 | 19.192 | 28.579 | – | – |
| | | ResNet34 | 97.834 | 25.060 | 32.863 | – | – |
| | | VGG-Small | 16.799 | 9.717 | 17.293 | – | – |
| Snapdragon 888 | 1 | ResNet18 | 77.205 | 15.547 | 16.111 | 123.618 | 25.240 |
| | | ResNet34 | 152.887 | 22.906 | 23.893 | 234.972 | 41.648 |
| | | VGG-Small | 25.133 | 2.410 | 2.543 | – | – |
| | 2 | ResNet18 | 46.297 | 19.309 | 19.321 | – | – |
| | | ResNet34 | 93.615 | 20.473 | 22.489 | – | – |
| | | VGG-Small | 16.001 | 1.920 | 2.213 | – | – |
| | 4 | ResNet18 | 33.524 | 13.699 | 14.332 | – | – |
| | | ResNet34 | 67.495 | 19.020 | 21.157 | – | – |
| | | VGG-Small | 11.743 | 2.882 | 2.768 | – | – |
| | 8 | ResNet18 | 33.761 | 26.108 | 58.989 | – | – |
| | | ResNet34 | 67.876 | 37.018 | 61.315 | – | – |
| | | VGG-Small | 11.752 | 27.615 | 16.774 | – | – |
| Raspberrypi 3B+ | 1 | ResNet18 | 740.509 | 158.732 | 175.256 | 1460.723 | 241.713 |
| | | ResNet34 | 1536.915 | 240.606 | 254.810 | 2774.888 | 435.170 |
| | | VGG-Small | 257.079 | 24.479 | 25.790 | – | – |
| | 2 | ResNet18 | 667.012 | 143.716 | 106.894 | – | – |
| | | ResNet34 | 933.149 | 144.287 | 158.868 | – | – |
| | | VGG-Small | 145.427 | 14.503 | 15.628 | – | – |
| | 4 | ResNet18 | 562.567 | 108.585 | 116.640 | – | – |
| | | ResNet34 | 976.223 | 159.258 | 183.698 | – | – |
| | | VGG-Small | 191.470 | 10.839 | 10.196 | – | – |
| | 8 | ResNet18 | 877.026 | 279.660 | 356.239 | – | – |
| | | ResNet34 | 1638.035 | 389.924 | 485.260 | – | – |
| | | VGG-Small | 399.338 | 110.448 | 142.978 | – | – |
| Raspberrypi 4B | 1 | ResNet18 | 448.744 | 80.822 | 82.380 | 688.838 | 120.348 |
| | | ResNet34 | 897.735 | 112.837 | 119.536 | 1362.893 | 209.276 |
| | | VGG-Small | 150.814 | 11.177 | 12.024 | – | – |
| | 2 | ResNet18 | 261.861 | 49.079 | 55.279 | – | – |
| | | ResNet34 | 525.735 | 67.480 | 79.468 | – | – |
| | | VGG-Small | 89.284 | 6.647 | 7.882 | – | – |
| | 4 | ResNet18 | 270.191 | 36.331 | 45.903 | – | – |
| | | ResNet34 | 572.423 | 53.866 | 70.841 | – | – |
| | | VGG-Small | 90.650 | 5.056 | 6.167 | – | – |
| | 8 | ResNet18 | 466.585 | 168.844 | 226.771 | – | – |
| | | ResNet34 | 879.375 | 264.638 | 319.789 | – | – |
| | | VGG-Small | 216.439 | 114.064 | 162.118 | – | – |

Table 16: Inference Efficiency on Hardware (4/4).

| Hardware | Threads | Arch. | Larq | | | daBNN | |
| --- | --- | --- | --- | --- | --- | --- | --- |
| | | | FP32 | BNN | ReAct | FP32 | BNN |
| Apple M1 | 1 | ResNet18 | 44.334 | 8.219 | 8.355 | – | – |
| | | ResNet34 | 88.334 | 12.505 | 12.771 | – | – |
| | | VGG-Small | 14.093 | 1.446 | 1.465 | – | – |
| | 2 | ResNet18 | 24.775 | 5.037 | 5.194 | – | – |
| | | ResNet34 | 47.179 | 7.425 | 7.690 | – | – |
| | | VGG-Small | 7.398 | 0.829 | 0.854 | – | – |
| | 4 | ResNet18 | 18.612 | 3.448 | 3.671 | – | – |
| | | ResNet34 | 27.515 | 4.965 | 5.254 | – | – |
| | | VGG-Small | 4.294 | 0.526 | 0.551 | – | – |
| | 8 | ResNet18 | 16.653 | 5.035 | 6.003 | – | – |
| | | ResNet34 | 27.680 | 6.445 | 6.953 | – | – |
| | | VGG-Small | 3.996 | 0.735 | 0.712 | – | – |
| | 16 | ResNet18 | 90.323 | 70.697 | 73.729 | – | – |
| | | ResNet34 | 162.057 | 130.907 | 125.362 | – | – |
| | | VGG-Small | 25.366 | 23.050 | 23.194 | – | – |
| Apple M1 Max | 1 | ResNet18 | 46.053 | 8.653 | 8.486 | – | – |
| | | ResNet34 | 91.861 | 12.593 | 13.039 | – | – |
| | | VGG-Small | 14.285 | 1.454 | 1.488 | – | – |
| | 2 | ResNet18 | 25.039 | 5.450 | 5.361 | – | – |
| | | ResNet34 | 51.860 | 7.579 | 8.925 | – | – |
| | | VGG-Small | 7.657 | 0.855 | 0.896 | – | – |
| | 4 | ResNet18 | 14.708 | 3.625 | 3.888 | – | – |
| | | ResNet34 | 27.933 | 5.266 | 6.021 | – | – |
| | | VGG-Small | 4.292 | 0.576 | 0.620 | – | – |
| | 8 | ResNet18 | 10.660 | 3.718 | 4.510 | – | – |
| | | ResNet34 | 18.988 | 4.745 | 5.457 | – | – |
| | | VGG-Small | 3.432 | 0.560 | 0.629 | – | – |
| | 16 | ResNet18 | 60.500 | 47.727 | 53.900 | – | – |
| | | ResNet34 | 120.449 | 91.464 | 96.356 | – | – |
| | | VGG-Small | 21.354 | 13.868 | 15.311 | – | – |

