# OpenReview forum: "BiBench: Benchmarking and Analyzing Network Binarization"
_ICLR.cc/2023/Conference — Submitted to ICLR 2023_

### Official Review · Reviewer_kQWB · 2022-10-21

**Confidence:** 4
**Correctness:** 3
**Technical Novelty And Significance:** 2
**Empirical Novelty And Significance:** 3
**Recommendation:** 5

**Clarity, Quality, Novelty And Reproducibility:**

The writing of this paper is very good, its easy to understand. As a benchmrah, its novelty is less important, its main contribution is that it can help newcoming methods to be more value. Its reproducibility needs the open code.

**Strength And Weaknesses:**

Strengths:
1. This paper provides the first benchmark in the area of binary algorithms. It consides the accuracy and efficiency of binary models. Specially, the authors explore the track of hardware inference, which is very meaningful.
2. With very much exiperiments, the paper analyzes the pros and cons of existing methods.

Weakness:
1. As a benchmark, more principles about binary algorithms are expected, such as its formula mode in mathmatics and softwares, its compatibility with current hardware.
2. Some abbreviation of letter are not explained. In the XNOR-Net of Appendix, sub-tensors in 'I', then what if the meaning of 'I'?
3. Because there are many experiments, many details are unclear in the paper. The open source code of this benchmark is very helpful to this area.

**Summary Of The Paper:**

This paper provides one benchmark named BiBench which is desifned to test the efficiency, robustness and generalization of binary algorithms. It mainly contains two parts, one for accuracy and another is for efficiency, and each part has several components. The authors build quantitative indicators to measure the binary algorithms, which is very significative. The BiBench is the first benchmark in the area of binary models.

**Summary Of The Review:**

The BiBench explores the advantages and disadvantages of different binary algorithms, and it shed a light to furture methods.

---

> ### Author Response · Authors · 2022-11-18
> **Response to Reviewer kQWB**
>
> We are deeply grateful for the reviewer’s support of our work, and we thank the reviewer for the constructive and helpful suggestions.
>
> >  **[Q1]** As a benchmark, more principles about binary algorithms are expected, such as its formula mode in mathmatics and softwares, its compatibility with current hardware.
>
> **[A1]** Thank you for pointing it out. We added more details about the formulas of binarization algorithms in Appendix A of our revision.
>
> During (software) training of a binarized model, the sign function is usually applied in the forward propagation. STE or other gradient approximations are applied in the backward propagation to make the binarized model trainable. The formulations of binarization can be expressed as
>
> $$\text{Forward:}\ \boldsymbol{b_x}=\operatorname{sign}(\boldsymbol{x})=\begin{cases}1& \text{if } \boldsymbol{x} \ge 0, \\\\
> -1& \text{otherwise }\end{cases},$$
>
> $$\text{Backward:}\ \boldsymbol{g_x}=\begin{cases}\boldsymbol{g}_{\operatorname{sign}(\boldsymbol x)}& \text{if } |\boldsymbol x| \leq 1\\\\ 0 & \text{otherwise }\end{cases},$$
>
> where $\boldsymbol{b_x}$ and $\boldsymbol{g_x}$ denote the binarized weight/activation and the corresponding gradient, respectively.
>
> For real-world hardware deployment, every 32 binarized parameters are packed together using 32-bit instructions and computed simultaneously, which is the main principle for acceleration. For compression of binary algorithms. Instructions including XNOR (or combine EOR and NOT) and Popcount enable binarized networks to deploy on real-world hardware. XNOR (exclusive-XOR) gate, a combination of an XOR gate followed by an inverter. XOR (also known as EOR) is a pervasive instruction that has long existed in assembly instructions for all target platforms. Popcount is short for “population count”, which counts the number of set bits in the 32-bit machine word. Most edge chips in use today include Popcount, such as ARM CPUs, X86 CPUs, FPGAs, etc. Therefore, binary neural networks are supported by mainstream edge devices.
>
> >  **[Q2]** Some abbreviation of letter are not explained. In the XNOR-Net of Appendix, sub-tensors in 'I', then what if the meaning of 'I'?
>
> **[A2]** We are sorry for the confusion. We corrected the relevant notation for XNOR-Net in Appendix A of our revision. The 'I' means the activation (input) tensor for XNOR-Net, which should be denoted as 'a' according to the context.
>
> XNOR-Net obtains the channel-wise scaling factors $\boldsymbol \alpha=\frac{\left\||\boldsymbol{w}\right\||}{\left|\boldsymbol{w}\right|}$ for the weight and $\boldsymbol{K}$ contains scaling factors $\beta$ for all sub-tensors in activation $\boldsymbol{a}$. We can approximate the convolution between activation $\boldsymbol{a}$ and weight $\boldsymbol{w}$ mainly using binary operations:
>
> $$\boldsymbol o = (\operatorname{sign}(\boldsymbol{a}) \circledast \operatorname{sign}(\boldsymbol{w})) \odot \boldsymbol{K} \boldsymbol \alpha,$$
>
> where $\boldsymbol{w} \in \mathbb{R}^{c \times w \times h}$ and $\boldsymbol{a} \in \mathbb{R}^{c \times w_{\text {in }} \times h_{\text {in }}}$ denote the weight and input tensor, respectively. And the STE is also applied in the backward propagation of the training process.
>
> >  **[Q3]** Because there are many experiments, many details are unclear in the paper. The open source code of this benchmark is very helpful to this area.
>
> **[A3]** Thanks for pointing it out. We revised our paper carefully and clarified some confusing contents in our revision. We also follow the reviewer's comments to release our first version code and corresponding documents in the supplementary material. We will improve this project continuously and publish it once upon acceptance.

---

> ### Author Response · Authors · 2022-11-24
> **Do you need further clarification or answers?**
>
> Thank you again for your comments, and we have revised our paper and released our code following all your suggestions in previous comments.
>
> We noticed there is a change to your score. We would like to ask if you have any other question? We are very willing to further answer and clarify for you.

---

### Official Review · Reviewer_Byew · 2022-10-24

**Confidence:** 5
**Correctness:** 3
**Technical Novelty And Significance:** 1
**Empirical Novelty And Significance:** 2
**Recommendation:** 3

**Clarity, Quality, Novelty And Reproducibility:**

The paper is easy to follow. Novelty is very limited. Reproducibility seems to be not a problem.

**Strength And Weaknesses:**

Although I very much agree with the motivation of this work, I also see very interesting settings and conclusions. However, I still have the following concerns:

- Why I think the evaluation of BNN on many downstream vision tasks does not make much sense now: The classification task model is usually used as the Backbone of other downstream tasks. For example, the imagenet pre-trained full precision ResNet backbone is widely used in a large number of vision tasks. Therefore, for a binary model with a serious accuracy degradation problem, it is of practical significance to first achieve sufficient accuracy on ImageNet, and then verify it in more downstream tasks. Therefore, the robustness of BNN does not seem to be an immediate concern until a feasible level of accuracy is reached.

-  There is certain doubt about the significance of the assessment. For example, the authors concluded that "There is no obvious difference in the theoretical complexity among binarization algorithms. ". I think this is mainly because similar architectural bases have been selected in the comparison. Many earlier methods such as XNOR, DoReFa, BiReal, xnor++ etc., are based on ResNet18 Backbone. More recent works selected in this paper as ReCU and FDA are based on the ReActNet network structure. So it is not surprising that there is no difference in theoretical complexity and inference time. Furthermore, the acc improvements of ReCU and FDA are about within 1% compared to ReActNet. Therefore, using the same Backbone with only marginal acc improvement brings little meaning to their benchmarking.
More BNN works should be included, especially those having different architectural designs, e.g., [1,2,3,4], etc.

- At present, the optimization of BNN has the problem of objective mismatch, and we usually need to use latent weight for training. Therefore, from the shape of the model parameters (only two states +1 and -1), the parameter update method (sign flipping), and the information flow path mode are significantly different from the fp model, why the author only considers the use of the full-precision model for the Architecture benchmarking? Why do the authors think that full-precision architectures are naturally suitable for BNN? In this work [3], the author verified that many full-precision model designs, such as bottleneck convolution, ConvBlock without shortcut connections (e.g., VGG, inceptionNet) are actually completely unsuitable for BNN.

- The results of the robustness analysis are nice. However, they are not new as well. [5,6] show that discrete BNNs exhibit superior stability and robustness against adversarial attacks.

- An important work [7] is missing in the training consumption track. [7] considers sign flipping without using the latent weight in the backward pass, which could significantly reduce the training epochs. The most compared optimizers are from the same type relying on the fp latent weights. Most of those proposed gradient approximation methods only got marginal contributions in their improvements. The most acc gains were from the architecture design.

- The inference speed test was originally the part I was most expected in this paper. However, as the authors said, extremely limited inference libraries make the contribution of this part minimal, and basically no new insights. Since most of the inference libs used by the authors have ceased development and maintenance, most of them do not have any performance optimizations for recent BNN models, so the evaluation results provided may not be the best performance. For example, ReActNet-A is based on the MobileNet-V1 Backbone. Neither larq nor DaBNN provide an implementation based on this Backbone and thus no corresponding instruction-set optimization and assembly optimization for the model on Arm. Without specific optimization, the implementation of a new model architecture usually does not have a significant efficiency gain.


[1] Zhang, Yichi, Zhiru Zhang, and Lukasz Lew. "PokeBNN: A Binary Pursuit of Lightweight Accuracy." Proceedings of the IEEE/CVF Conference on Computer Vision and Pattern Recognition. 2022.

[2] Bethge, Joseph, et al. "Meliusnet: Can binary neural networks achieve mobilenet-level accuracy?." arXiv preprint arXiv:2001.05936 (2020).

[3] Bethge, J., Yang, H., Bornstein, M., & Meinel, C. (2019). Back to simplicity: How to train accurate bnns from scratch?. arXiv preprint arXiv:1906.08637.

[4] Martinez, B., Yang, J., Bulat, A., & Tzimiropoulos, G. (2020). Training binary neural networks with real-to-binary convolutions. arXiv preprint arXiv:2003.11535.

[5] A. Galloway et al. Attacking binarized neural networks. arXiv:1711.00449, 2017.

[6] E. Khalil et al. Combinatorial attacks on binarized neural networks. arXiv:1810.03538, 2018.

[7] Helwegen, Koen, et al. "Latent weights do not exist: Rethinking binarized neural network optimization." Advances in neural information processing systems 32 (2019).

**Summary Of The Paper:**

This paper presents a benchmark for binary neural networks. The perspectives of the proposed benchmark include BNN methods, architectures, multiple tasks, and inference tests on various hardware. The authors summarize some insights and offer practical guidance.

**Summary Of The Review:**

Rather than saying that this is a benchmark paper, it is actually more like evaluating some selected BNNs from different perspectives. The author's original intention is constructive. However, the selected work has certain limitations, with a large number of ancient BNNs, some newer architecture designs, and optimization methods that are missing. In addition, it is not very reasonable to directly use the architectures of the fp-network for architecture benchmarking. The inference speed tests are greatly limited by the libs used, and their results provide very limited guidance. It is difficult for me to get a clear takeaway from a lot of experiments and data.

---

> ### Author Response · Authors · 2022-11-18
> **Response to Reviewer Byew (1/4)**
>
> We thank the reviewer for the detailed feedback and comments:
>
> >  **[Q1]** Why I think the evaluation of BNN on many downstream vision tasks does not make much sense now: The classification task model is usually used as the Backbone of other downstream tasks. For example, the imagenet pre-trained full precision ResNet backbone is widely used in a large number of vision tasks. Therefore, for a binary model with a serious accuracy degradation problem, it is of practical significance to first achieve sufficient accuracy on ImageNet, and then verify it in more downstream tasks. Therefore, the robustness of BNN does not seem to be an immediate concern until a feasible level of accuracy is reached.
>
> **[A1]** First, except that Pascal VOC and COCO are downstream vision tasks (2D image detection), most evaluations in BiBench are performed on basic tasks in various fields (rather than downstream vision tasks), including 2D image classification tasks on CIFAR10, ImageNet, CIFAR10-C datasets, 3D visual classification task on ModelNet40 dataset, language understanding tasks on GLUE dataset, and keyword spotting classification task on Google Speech Commands dataset. We emphasize that network binarization is not a technology that is only attached to vision; conversely, it draws wide attention in various deep learning fields and thus should be evaluated comprehensively. For example, our benchmark shows that although there is much room for improvement on ImageNet, the binarization can achieve less than a 2% drop on the 3D vision task ModelNet40 (accuracy in Table 6 of revision: PointNet_vanilla XNOR++ 85.66% vs. FP32 86.80%) and also less than a 4% drop on Speech Commands (accuracy in Table 7: DFSMN FDA 93.91% vs. FP32 97.51%). These results prove that the development and applicability of network binarization in different learning fields are different. Thus, measuring network binarization performance on different modality tasks is necessary. The extensive evaluation of various learning tasks also makes the conclusions of our BiBench sufficiently reliable and referable.
>
> Furthermore, we would like to clarify that it is urgent and necessary to benchmark network binarization on downstream vision tasks. As one of the most typical downstream vision tasks, object detection tasks are considered by some specific binary networks [1,2,3], so the evaluation of general binarization algorithms (such as the selected algorithms in BiBench) on object detection tasks with the standard training pipeline is necessary. The evaluation allows researchers to establish reasonable baselines for these detection tasks in future studies. Our evaluation also shows that the performance of upstream ImageNet and downstream COCO tasks are not completely aligned for different binarization algorithms, e.g., ReCU outperforms ReActNet by 1.84% on COCO while suffering a 0.66% drop against ReActNet on ImageNet, and similar phenomena also exist between XNOR vs. DoReFa and other binarization algorithms。 These differences should not be completely ignored.
>
> [1] Wang Z, Wu Z, Lu J, et al. BiDet: An efficient binarized object detector. CVPR, 2020.
>
> [2] Xu S, Zhao J, Lu J, et al. Layer-wise searching for 1-bit detectors. CVPR, 2021.
>
> [3] Tu Z, Chen X, Ren P, et al. Adabin: Improving binary neural networks with adaptive binary sets. ECCV, 2022.

---

> > ### Author Response · Authors · 2022-11-18
> > **Response to Reviewer Byew (2/4)**
> >
> > >  **[Q2]** There is certain doubt about the significance of the assessment. For example, the authors concluded that "There is no obvious difference in the theoretical complexity among binarization algorithms. ". I think this is mainly because similar architectural bases have been selected in the comparison. Many earlier methods such as XNOR, DoReFa, BiReal, xnor++ etc., are based on ResNet18 Backbone. More recent works selected in this paper as ReCU and FDA are based on the ReActNet network structure. So it is not surprising that there is no difference in theoretical complexity and inference time. Furthermore, the acc improvements of ReCU and FDA are about within 1% compared to ReActNet. Therefore, using the same Backbone with only marginal acc improvement brings little meaning to their benchmarking. More BNN works should be included, especially those having different architectural designs, e.g., [1,2,3,4], etc.
> >
> > **[A2]** First, our BiBench aims to evaluate generic binarization algorithms at the operator level, so our evaluation of the algorithm is decoupled from the neural architecture. As we state in Section 2.1, since the fundamental difference between binarized and full-precision models is that the former applies binarized operators, our BiBench selects 8 generic binarization algorithms focusing on operator improvement while excluding the algorithms and techniques requiring specified local structures or training pipelines. Therefore, the binarization algorithms evaluated in our BiBench are generic to neural architectures (including the mentioned ReActNet architecture), i.e., these architectures are orthogonal to various evaluated binarized operators. For example, for ReActNet, we only keep the technique that affects the inside of the operator (RSign) and exclude the techniques related to local structure (duplicate activation) since it is hard to transfer across all architectures like transformers. For Bi-Real Net, additional shortcuts are not adopted. And following the reviewer's suggestion, we evaluated ResNet-Bi-Real-18 and MobileNet-ReActNet-A as a separate architecture to compare their CIFAR-10 accuracy:
> >
> > |                                          | **FP32** | **BNN** | **XNOR** | **DoReFa** | **Bi-Real** | **XNOR++** | **ReActNet** | **ReCU** | **FDA** |
> > |------------------------------------------|----------|---------|----------|------------|-------------|------------|--------------|----------|---------|
> > | Overall CNNs in BiBench (Overall_Metric) |          | 72.9    | 83.73    | 83.86      | 85.02       | 78.95      | 86.2         | 83.5     | 86.34   |
> > | ResNet-Bi-Real-18 (Acc %)                | 94.98    | 89.84   | 91.51    | 89.63      | 91.76       | 89.55      | 91.55        | 92.94    | 90.81   |
> > | MobileNet-ReActNet-A (Acc %)             | 89.08    | 86.45   | 85.01    | 86.38      | 86.25       | 86.26      | 86.25        | 86.14    | 86.29   |
> >
> > Despite improved performance over similar original architectures (e.g., ResNet-18 vs. ResNet-Bi-Real-18), the accuracy for MobileNet-ReActNet-A and ResNet-Bi-Real-18 with different sizes and structures of various binarization algorithms still presents a large difference due to binarized operators' differences. And the algorithms also perform similarly to the overall results of CNN neural architectures in BiBench (Table 2), demonstrating that the evaluation results of our BiBench are reliable and generalizable to the corresponding neural architectures.
> >
> > We also emphasize that the neural architectures are the same when comparing all 8 binarization algorithms. We only change and compare different binarized operators in each architecture to control the variables. So there are no additional structures in evaluations of ReActNet, ReCU, and FDA algorithms, but the architectures are aligned with full-precision structures for a fair evaluation. Moreover, even under the same architecture, the accuracy difference brought by different binarized operators is still significant. For example, on the widely studied basic CIFAR-10 classification task, the gap between the best and worst results among binarized operators is still 2.2% (metric in Table 2: XNOR++ 94.52% vs. ReCU 96.72%), and the gap even as high as 32.97% on ShapeNet task (XNOR 73.62% vs. ReCU 40.65%). In addition, when we put the perspective in other tracks, the difference between these operators is even greater. For example, in the corruption robustness track, there is a huge gap between DoReFa and ReCU, and the difference in the training consumption track is also significant. These phenomena also prove that it is necessary to benchmark various binarization algorithms from the operator's perspective because their performance differences under various tracks are obvious.

---

> > > ### Author Response · Authors · 2022-11-18
> > > **Response to Reviewer Byew (3/4)**
> > >
> > > >  **[Q3]** At present, the optimization of BNN has the problem of objective mismatch, and we usually need to use latent weight for training. Therefore, from the shape of the model parameters (only two states +1 and -1), the parameter update method (sign flipping), and the information flow path mode are significantly different from the fp model, why the author only considers the use of the full-precision model for the Architecture benchmarking? Why do the authors think that full-precision architectures are naturally suitable for BNN? In this work [3], the author verified that many full-precision model designs, such as bottleneck convolution, ConvBlock without shortcut connections (e.g., VGG, inceptionNet) are actually completely unsuitable for BNN.
> > >
> > > **[A3]** First, as we emphasized, BiBench focuses on benchmarking operators of different binarization algorithms, and neural architectures are treated as a separate track to evaluate binarization's effect on various architectures comprehensively. Comparing the full-precision and binarized networks of the aligned architecture helps us intuitively assess the capabilities of these binarization algorithms. At the same time, the previously mentioned structural variants, such as Bi-Real Net and ReActNet, can be viewed as a separate architecture to be evaluated in our BiBench with the same full-precision and binarized versions.
> > >
> > > Moreover, there is no mention of "full-precision architectures are naturally suitable for BNN" in our manuscript. On the contrary, some structures in certain neural architectures should be replaced with binarization-friendly versions, or they cannot converge, such as the aggregator in the PointNet and attention in BERTs. We further clarify this issue in Section 3.1 and Appendix C. Our selected neural architectures and optimization methods are widely adopted and recognized in previous research, which allows the BiBench to be a consistent and fair binarization benchmark.
> > >
> > > >  **[Q4]** The results of the robustness analysis are nice. However, they are not new as well. [5,6] show that discrete BNNs exhibit superior stability and robustness against adversarial attacks.
> > > [5] A. Galloway et al. Attacking binarized neural networks. arXiv:1711.00449, 2017.
> > > [6] E. Khalil et al. Combinatorial attacks on binarized neural networks. arXiv:1810.03538, 2018.
> > >
> > > **[A4]** We should state that the "corruption" evaluated by our BiBench and the "adversarial attacks" mentioned by the reviewer belong to different scopes in the research field of robustness. Specifically, adversarial robustness is a good measure of a model's worst-case performance. However, it can not reflect a model's robustness to common corruption in the natural world [1,2], which is precisely what corruption research focuses on. To our knowledge, BiBench is the first work to evaluate the corruption robustness of network binarization algorithms. And we also clearly explained the reason for evaluating the corruption robustness of network binarization in Section 3, i.e., the robustness of binarization on deployment is important to deal with bad cases like perceptual device damage, which is a common problem with low-cost equipment in real-world deployments.
> > >
> > > Moreover, except focusing on different research scopes, our BiBench evaluates the robustness of various representative binarization algorithms rather than just a certain one, which makes our conclusions of corruption robustness enlightening to the future works of researchers and makes our work different from existing studies.
> > >
> > > [1] Ren J, Pan L, Liu Z. Benchmarking and analyzing point cloud classification under corruptions. ICLR, 2022.
> > >
> > > [2] Hendrycks D, Dietterich T. Benchmarking Neural Network Robustness to Common Corruptions and Perturbations. ICLR, 2018.

---

> > > > ### Author Response · Authors · 2022-11-18
> > > > **Response to Reviewer Byew (4/4)**
> > > >
> > > > >  **[Q5]** An important work [7] is missing in the training consumption track. [7] considers sign flipping without using the latent weight in the backward pass, which could significantly reduce the training epochs. The most compared optimizers are from the same type relying on the fp latent weights. Most of those proposed gradient approximation methods only got marginal contributions in their improvements. The most acc gains were from the architecture design.
> > > > [7] Helwegen, Koen, et al. "Latent weights do not exist: Rethinking binarized neural network optimization." Advances in neural information processing systems 32 (2019).
> > > >
> > > > **[A5]** We add related discussions about [7] in the revision.
> > > >
> > > > This work presents an optimization method forcibly bounding binarized operators without using the latent weight, but the evaluations in BiBench fix the operator and change the architectures or optimizers, etc. Since the binarized operators and optimization form in [7] cannot be decoupled, this algorithm cannot complete all evaluations in our BiBench and be fairly compared with other binarized operators. And the SGD and Adam optimizers evaluated in our BiBench are widely adopted by various classical and recent binarization works (including the original works of all 8 binarization algorithms in BiBench) and are thus representative.
> > > >
> > > > In addition, the results in BiBench show that the gradient approximation method brings significant improvements, and the improvement is orthogonal to the neural architectural track. For example, the binarized operators of DoReFa and Bi-Real differ only in the gradient approximation technique in BiBench (while the forward computation is the same). However, the latter has significant improvements over the former on Pascal VOC and Speech Commands tasks by 3.31% and 5.78%, respectively. The FDA algorithm also proposes a novel gradient approximation as its main contribution, bringing significant improvements and performing outstanding under multiple tracks. Another great advantage of gradient approximation methods is that additional computation and expensive library adaptation are not required when deployment inference, so the contribution of such techniques should not be ignored.
> > > >
> > > > >  **[Q6]** The inference speed test was originally the part I was most expected in this paper. However, as the authors said, extremely limited inference libraries make the contribution of this part minimal, and basically no new insights. Since most of the inference libs used by the authors have ceased development and maintenance, most of them do not have any performance optimizations for recent BNN models, so the evaluation results provided may not be the best performance. For example, ReActNet-A is based on the MobileNet-V1 Backbone. Neither larq nor DaBNN provide an implementation based on this Backbone, and thus no corresponding instruction-set optimization and assembly optimization for the model on Arm. Without specific optimization, the implementation of a new model architecture usually does not have a significant efficiency gain.
> > > >
> > > > **[A6]** In BiBench, we strive to collect all existing deployment libraries that can sufficiently support binarization algorithms (Larq and daBNN), which should be open-source and have an easy-to-use deployment pipeline. For any binarization researcher, these deployment libraries are almost all their choices for a fair performance comparison of binarization algorithms on real-world edge hardware. Based on these deployment libraries, we conduct a large number of edge hardware evaluations of representative binarization algorithms, which help to demonstrate the capability of existing binarization algorithms when they land.
> > > >
> > > > Another non-negligible contribution of our hardware inference track is that we reveal that binarization is a compression technology born for the edge based on massive evaluation results of edge devices with different computing power (as Figure 3). As we presented in Section 5.2.3, the lower the chip's computing power, the higher speedup achieved. After deploying and evaluating binarized models across dozens of chips, we compare each chip's average speedup of the binarization algorithms. A counterintuitive finding is that the higher the chip capability, the worse the speedup of binarization on the chip. Further observations showed that the contradiction is mainly because higher-performance chips have more acceleration brought by multi-threading when running floating-point models; thus, the speedup of binarized models is relatively slow. This means that the scenarios where binarization technology comes into play are edge chips with low performance and cost. The vast majority of binarization methods have almost identical inference performance, and the mean-shifting operation of ReActNet on activation slightly affects the efficiency, i.e., binarized models must satisfy fixed deployment paradigms and have almost identical performance.

---

### Official Review · Reviewer_LQy6 · 2022-11-03

**Confidence:** 3
**Correctness:** 2
**Technical Novelty And Significance:** 3
**Empirical Novelty And Significance:** 3
**Recommendation:** 5

**Clarity, Quality, Novelty And Reproducibility:**

Although the paper proposes a novel benchmark, the clarity of the explanation is not sufficient (see W2). Moreover, a number of typos and bad sentences render the reading even more strenuous. In terms of reproducibility, I could not find the code. For a benchmark, the presence of well-documented and easy-to-use code is paramount.

**Strength And Weaknesses:**

Strengths:

(S1) The paper sheds a light on important issues related to network binarization, such as robustness and empirical time improvement.

(S2) The benchmark is extensive and covers a number of architectures, learning tasks, hardware, and robustness evaluation.

(S3) The paper evaluates 8 different network binarization methods, from older to more recent ones.

(S4) The proposed benchmark BiBench is useful for the ML community.

Weaknesses:

(W1) The paper is poorly written: there is a large number of typos, broken sentences and bad use of English. The paper would benefit from thorough proofreading and rewriting. Some sentences make no sense or are hard to understand. For reference, here is an incomplete list of sentences and typos:

- severe accuracy challenges **but** diminishing
- 1 corruption benchmarks
- pushing network binarization research to be accurate and efficient
- Binarization ~~technology~~ compresses
- to fully exert the generic of binarization technology.
- We train the 1× number of training epochs
- which requires specifically studied in binarization research.

(W2) Clarity: The paper should introduce the concepts more smoothly and walk the reader through them. Many concepts are assumed to be known and dropped without reference. The paper is therefore hard to read. Examples of unclear/unexplained concepts or claims are:

- “The most aggressive quantization technology” according to whom?
- “imaging modality task”: what is that? how is it defined?
- the 1-bit specialization of quantization” - missing reference
- “requiring specified local structures” such as?
- “model is exported in the ONNX” → Is this a common format? Where does it come from?
- What do the colours in Table 2-3 represent?

(W3) The paper would benefit from more rigour. E.g.,

- Popcount is never defined
- Examples of $\alpha$ and $w$ should be provided
- “The quadratic mean form is uniformly applied in BiBench to unify all metrics.” Why is this a good choice?

(W3) The intuition behind the evaluation metrics in Eq. 2 - 9 is unclear. Why not consider standard deviation as well?

(W4) The related work analysis should be more extensive and explain what are the choices determining the current selection of algorithms. A concurrent work [1] presents other models for binarization in its related work section. How are the algorithms in Table 1 chosen? Why are they representative?

[1] Shang, Y., Xu, D., Zong, Z. and Yan, Y., 2022. Network Binarization via Contrastive Learning. ECCV

(W5) As a benchmark, it should probably compare with other, simpler strategies to make the network more compact. For instance, one such strategy could be dropout or model quantization. There is no need to be exhaustive there, but there should be the possibility for researchers working on Network binarization to assess their methods on more traditional techniques.

(W5) It is not clear to what extent the chosen datasets and tasks are challenging or representative. The paper should elaborate more on why some of the tasks have been chosen. The description in Section 3.1 assumes the reader knows the tasks but does not provide additional information on why they are representative.

(W6) Code: The implementation is not available. A benchmark should provide the code (in this case anonymous) for reproducibility. Moreover, the code has to be clear, well-documented, and easy to run.

**Summary Of The Paper:**

The paper proposes a benchmark to evaluate Network binarization. Network binarization is a compression method for neural networks that transforms the layers into binary vectors. Given the lack of a comprehensive evaluation and benchmarking methodology, the paper proposes sets of tasks, measures, hardware, and methods to validate new algorithms. Finally, the paper shows how network binarization is not a method that can seemingly apply to any network with no tuning.

**Summary Of The Review:**

Although the paper is interesting and the problem relevant, the presentation is sloppy and most of the decisions and claims not fully motivated. For this reason, I believe the paper is not ready for publication at ICLR.

---

> ### Author Response · Authors · 2022-11-18
> **Response to Reviewer LQy6 (1/5)**
>
> We thank the reviewer for the detailed feedback and comments. We respond to the concerns below:
>
> >  **[Q1]** The paper is poorly written: there is a large number of typos, broken sentences and bad use of English. The paper would benefit from thorough proofreading and rewriting. Some sentences make no sense or are hard to understand.
>
> **[A1]** Thanks for pointing it out. In the revision, we followed the reviewer's comments and carefully revised the whole manuscript to address the issues mentioned, and other typos, broken sentences, and bad use of English.
>
> >  **[Q2]** Clarity: The paper should introduce the concepts more smoothly and walk the reader through them. Many concepts are assumed to be known and dropped without reference. The paper is therefore hard to read. Examples of unclear/unexplained concepts or claims are:
>
> **[A2]** Thanks for your comments. In the revision, we clarify the concepts and claims existing in our manuscript in detail. And the concepts and claims mentioned in the comments are clarified as follows:
>
> >  **[Q2.1]** “The most aggressive quantization technology” according to whom?
>
> **[A2.1]** We clarified it in Section 1 and Appendix A of our revision.
>
> In previous studies, quantization schemes with lower bit-widths were regarded as more aggressive schemes [1,2,3,4] because lower bit-widths usually lead to higher compression and speedup but result in more Severe accuracy loss. With the lowest bit-width among all quantization approaches, 1-bit quantization (binarization) is regarded as the most aggressive quantization technique [4], facing severe challenges in terms of accuracy but enjoying the highest compression and speedup ratios.
>
> [1] Rusci M, Capotondi A, Benini L. Memory-driven mixed low precision quantization for enabling deep network inference on microcontrollers. MLSys, 2020.
>
> [2] Choukroun Y, Kravchik E, Yang F, et al. Low-bit quantization of neural networks for efficient inference. ICCVW, 2019.
>
> [3] https://intellabs.github.io/distiller/quantization.html#aggressive-quantization-int4-and-lower
>
> [4] Qin H, Ma X, Ding Y, et al. BiFSMN: Binary Neural Network for Keyword Spotting. IJCAI, 2022.
>
>
> > **[Q2.2]** “imaging modality task”: what is that? how is it defined?
>
> **[A2.2]** We are sorry for the confusion and clarified it in Section 1 of the revision.
>
> The "imaging modality task" in our original manuscript would refer to tasks with 2D image modality input, including image classification (CIFAR10, ImageNet, and CIFAR10-C) and object detection (Pascal VOC and COCO) tasks. The statement is revised to "learning tasks with image modality inputs" to avoid confusion.
>
>
> >  **[Q2.3]** the 1-bit specialization of quantization” - missing reference
>
> **[A2.3]** We added the relevant reference [1,2] to the revision (Section 1) and further clarified this claim in Appendix A.
>
> Since the parameters are quantized to binary, network binarization approaches usually use a simple sign function as the quantizer instead of directly sharing the quantizer with multi-bit (2-8 bit) quantization [1,2]. Specifically, as [1] describes, for multi-bit uniform quantization, given the bit width $b$ and the floating-point activation/weight $x$ following in the range $(l, u)$, the complete quantization-dequantization process of uniform quantization can be defined as
>
> $$Q_U (x) = round(x/\Delta)\Delta,$$
>
> where the original range $(l, u)$ is divided into $2^b−1$ intervals $Pi, i \in (0, 1, \cdots, 2^b−1)$, and $\Delta = (u−l)/(2^b−1)$ is the interval length. When $b=1$, the $Q_U (x)$ equals the sign function, and the binary function is expressed as
>
> $$QB(x) = sign(x).$$
>
> Therefore, binarization can be regarded as the 1-bit specialization of quantization.
>
> [1] Gong R, Liu X, Jiang S, et al. Differentiable soft quantization: Bridging full-precision and low-bit neural networks. ICCV, 2019.
>
> [2] Gholami A, Kim S, Dong Z, et al. A Survey of Quantization Methods for Efficient Neural Network Inference. Low-Power Computer Vision, 2021.
>
> >  **[Q2.4]** “requiring specified local structures” such as?
>
> **[A2.4]** We clarified it in Section 2.1 and Appendix A of our revision.
>
> In the original manuscript, the techniques that "requiring specified local structures" refers to techniques only customized for certain architectures, such as the bi-real shortcut for ResNet architectures (CNNs) in Bi-Real Net [1] and duplicate activation in ReActNet [2] for ResNet and MobileNet architectures (CNNs). As stated in Section 2.1 of the manuscript, we exclude these techniques for a comprehensive and fair evaluation of network binarization and focus on benchmarking binarized operators of different binarization methods.
>
> [1] Liu Z, Shen Z, Savvides M, et al. Reactnet: Towards precise binary neural network with generalized activation functions. ECCV, 2020.
>
> [2] Liu Z, Wu B, Luo W, et al. Bi-real net: Enhancing the performance of 1-bit cnns with improved representational capability and advanced training algorithm. ECCV, 2018.

---

> > ### Author Response · Authors · 2022-11-18
> > **Response to Reviewer LQy6 (2/5)**
> >
> > >  **[Q2.5]** “model is exported in the ONNX” → Is this a common format? Where does it come from?
> >
> > **[A2.5]** We further clarified the description of ONNX (Open Neural Network Exchange) and added related references in Section 4 and Appendix A.
> >
> > As their official website describes ( https://onnx.ai/ ), ONNX is an open format built to represent machine learning models. ONNX defines a common set of operators - the building blocks of machine learning and deep learning models - and a common file format to enable AI developers to use models with various frameworks, tools, runtimes, and compilers. And the existing deployment libraries supporting binarization, such as Larq [1] and daBNN [2]. After converting the saved binarized models to ONNX format, they are converted to deployable formats by libraries.
> >
> > [1] Geiger L, Team P. Larq: An open-source library for training binarized neural networks. Journal of Open Source Software, 2020.
> >
> > [2] Zhang J, Pan Y, Yao T, et al. dabnn: A super fast inference framework for binary neural networks on arm devices. ACM MM, 2019.
> >
> > >  **[Q2.6]** What do the colours in Table 2-3 represent?
> >
> > **[A2.6]** We clarified it in the captions of Table 2 and 3.
> >
> > In Table 2 and 3,  the blue and red colors denote the best and worst in the row, respectively. For the overall metrics of each track, the colors are darker.
> >
> > >  **[Q3]** The paper would benefit from more rigour.
> >
> > **[A3]** Thanks for pointing it out. We carefully refined our manuscript, and below, we clarify the raised questions:
> >
> > > **[Q3.1]** Popcount is never defined
> >
> > **[A3.1]** We define and describe popcount and XNOR instructions in Section 2.1 and Appendix A.
> > The popcount instruction means Population Count per byte. This instruction counts the number of bits with one value in each vector element in the source register, places the result into a vector, and writes the vector to the destination register [1]. This instruction is applied to accelerate the inference of binarized networks [2,3] and is widely supported by various hardware, e.g., the definitions of popcount in ARM and x86 are in [1] and [4], respectively.
> >
> > [1] Arm A64 Instruction Set Architecture. https://developer.arm.com/documentation/ddi0596/2020-12/SIMD-FP-Instructions/CNT--Population-Count-per-byte- .
> >
> > [2] Hubara I, Courbariaux M, Soudry D, et al. Binarized neural networks. NeurIPS, 2016.
> >
> > [3] Rastegari M, Ordonez V, Redmon J, et al. Xnor-net: Imagenet classification using binary convolutional neural networks. ECCV, 2016.
> >
> > [4] AMD64 Architecture Programmer’s Manual Volume 3. https://www.amd.com/system/files/TechDocs/24594.pdf .
> >
> > >  **[Q3.2]** Examples of \alpha and w should be provided
> >
> > **[A3.2]** We follow the reviewer's comments to show the definitions and examples of $\alpha$ and $w$.
> >
> > As the definition in previous binarization works [1,2,3], the $w \in \mathbb{R}^{c_{in}×c_{out}×k×k}$ denotes real-value weight filter and $\alpha \in \mathbb{R}^{c_\text{out}}$ denotes the scaling factor ($c_\text{in}$, $k$, and $c_{out}$ denote input channel, kernel size, and output channel, respectively), and the $\alpha$ is calculated as $\alpha=\frac{\|w\|}{n}$.
> >
> > [1] Rastegari M, Ordonez V, Redmon J, et al. Xnor-net: Imagenet classification using binary convolutional neural networks. ECCV, 2016.
> >
> > [2] Liu Z, Shen Z, Savvides M, et al. Reactnet: Towards precise binary neural network with generalized activation functions. ECCV, 2020.
> >
> > [3] Qin H, Gong R, Liu X, et al. Binary neural networks: A survey. Pattern Recognition, 2020.
> >
> > >  **[Q3.3]** “The quadratic mean form is uniformly applied in BiBench to unify all metrics.” Why is this a good choice?
> >
> > **[A3.3]** We clarify the motivation for selecting the metric form in Section 3.1 and Appendix B of the revision.
> > The metrics in BiBench are proposed to measure the overall performance of binarization algorithms on each track. Compared to the most commonly used mean form of directly averaging each item, the metric of the quadratic mean form gives greater weight to larger items in a set, preventing the metrics from being unduly influenced by certain poorly performing items.

---

> > > ### Author Response · Authors · 2022-11-18
> > > **Response to Reviewer LQy6 (3/5)**
> > >
> > > >  **[Q4]** The intuition behind the evaluation metrics in Eq. 2 - 9 is unclear. Why not consider standard deviation as well?
> > >
> > > **[A4]** We further clarify the intuition of the metric. First, the design of all metrics follows the following rules: (1) the range of metrics is normalized to the interval [0, 1] for fair and easy comparison, (2) and the larger values of metrics mean higher performance. And the motivation for the selection of metric form is presented in A3.3.
> > >
> > > Moreover, compared to the quadratic mean, the standard deviation of the items in each track is hard to directly reflect the properties of different binarization algorithms in the evaluation. Even the metrics of full-precision models are quite different for most defined metrics. For example, for the learning task track, the full-precision models' accuracy varies greatly for different tasks (e.g., FP32 ResNet-18 CIFAR-10 94.82% vs. ImageNet 69.90%). Therefore, the standard deviation metrics of binarization algorithms are hard to show the performance of algorithms intuitively. As for the hyperparameter subterm in the training consumption track (Eq. (6)), we believe these data share more similar properties. Hence, the standard deviation is used here to measure the fluctuation of $A_{hyper}$ under various training settings.
> > >
> > > >  **[Q5]** The related work analysis should be more extensive and explain what are the choices determining the current selection of algorithms. A concurrent work [1] presents other models for binarization in its related work section. How are the algorithms in Table 1 chosen? Why are they representative?
> > > [1] Shang, Y., Xu, D., Zong, Z. and Yan, Y., 2022. Network Binarization via Contrastive Learning. ECCV
> > >
> > > **[A5]** In Section 1 and Appendix A of our revision, we analyze the binarization algorithms mentioned in [1]’s related work section (BNN, XNOR-Net, XNOR-Net++, Bi-Real Net, ProxyBNN, and ReActNet) and more related works we considered, and also clarify the rules determining the current selection of algorithms.
> > >
> > > First, we state that we obey some general rules for selecting binarization algorithms, i.e, the selected binarization algorithms should improve the binarized operator since it is the fundamental difference between binarized and full-precision models (as discussed in Section 2.1). And we thus also exclude the algorithms and techniques requiring specified local structures or training pipelines for a fair comparison.
> > >
> > > Then, we explained in detail the choice of binarization algorithms and why they are representative. When we built the BiBench, we considered various binarization algorithms with improved operator techniques, and now we list them in detail in the table below. We consider from the following perspectives, the purposes are making the selected binarization algorithms representative and should complete all evaluations in BiBench fairly: Operator Techniques (Yes/No), Year, Conference, Citation (to 2022/11/08), Open source (Yes/No), and Specified Structure / Training-pipeline (Yes/No/Optional).
> > >
> > > (1) We analyze the techniques proposed in these works. Following the general rules we mentioned, all considered binarization algorithms should have significant contributions to the improvement of the binarization operator (Operator Techniques: Yes) and should not include techniques that are bound to specific architectures and training pipelines to complete well all the evaluations of the learning task, neural architecture, and training consumption tracks in BiBench (Specified Structure / Training-pipeline: No/Optional, Optional means the techniques are included but can be decoupled with binarized operator totally).
> > >
> > > (2) We also consider the impact and reproducibility of these works. We prioritized the selection of works with more than 100 citations, which means they are more discussed and compared in binarization research and thus have higher impacts. Works in 2021 and later are regarded as the SoTA binarization algorithms and prioritized. Furthermore, we hope the selected works have official open-source implementations for reproducibility.
> > >
> > > Based on the above selections, eight binarization algorithms, i.e., BNN, XNOR-Net, DoReFa-Net, Bi-Real Net, XNOR-Net++, ReActNet, FDA, and ReCU, stand out and are fully evaluated by our BiBench.

---

> > > > ### Author Response · Authors · 2022-11-18
> > > > **Response to Reviewer LQy6 (4/5)**
> > > >
> > > > Table 5. Considered binarization algorithms and our final selections in BiBench. Bold means that the algorithm has an advantage in that column.
> > > >
> > > > **Algorithm** | **Operator Techniques** | **Specified Structure / Training-pipeline** | **Year** | **Conference** | **Citation (to 2022/11/08)** | Open source
> > > > ---------------|-------------------------|---------------------------------------------|----------|----------------|------------------------------|------------------
> > > >  **BNN**           | **Yes**                     | **No**                                          | 2016     | NeurIPS        | **1846**                         |  **Yes**
> > > >  **XNOR-Net**      | **Yes**                     | **No**                                          | 2016     | ECCV           | **4313**                         |  **Yes**
> > > >  **DoReFa-Net**    | **Yes**                     | **No**                                          | 2016     | ArXiv          | **174**                          |  **Yes**
> > > >  **Bi-Real Net**   | **Yes**                     |  **Optional**                                    | 2018     | ECCV           | **371**                          |  **Yes**
> > > >  CI-BCNN       | **Yes**                     | Yes                                         | 2019     | CVPR           | 77                           | **Yes**
> > > >  **XNOR-Net++**    | **Yes**                     | **No**                                          | 2019     | BMVC           | **118**                          |  **Yes**
> > > >  RBNN          | **Yes**                     | **No**                                          | 2020     | NeurIPS        | 62                           | **Yes**
> > > >  ProxyBNN      | **Yes**                     | Yes                                         | 2020     | ECCV           | 14                           | No
> > > >  **ReActNet**      | **Yes**                     | **Optional**                                    | 2020     | ECCV           | **151**                          |  **Yes**
> > > >  Si-BNN        | **Yes**                     | **No**                                          | 2020     | AAAI           | 24                           | No
> > > >  **FDA**           | **Yes**                     | **No**                                          | **2021**     | NeurIPS        | 11                           |  **Yes**
> > > >  **ReCU**          | **Yes**                     | **No**                                          | **2021**     | ICCV           | 15                           |  **Yes**
> > > >  LCR-BNN       | **Yes**                     | Yes                                         | **2022**     | ECCV           | 0                            | **Yes**
> > > >
> > > > >  **[Q6]** As a benchmark, it should probably compare with other, simpler strategies to make the network more compact. For instance, one such strategy could be dropout or model quantization. There is no need to be exhaustive there, but there should be the possibility for researchers working on Network binarization to assess their methods on more traditional techniques.
> > > >
> > > > **[A6]** Thanks for your constructive suggestions. In Section 1 and Appendix A of our revision, we compare the characteristics of different compression techniques including network binarization, multi-bit quantization, pruning (dropout), etc., which allows researchers to understand the applicable scenarios and advantages of binarization.
> > > >
> > > > Most existing network compression technologies aim to reduce the size and computation of full-precision models. Specifically, knowledge distillation supervises compact small (student) models by intermediate features and/or soft outputs of the large (teacher) model; model pruning and low-rank decomposition reduces network parameters and computation by pruning and low-rank approximation; compact model design directly designs a compact model. Although these compression technologies can effectively reduce the number of parameters, the compressed model still uses 32-bit floating-point numbers, which leaves room for further compression using model quantization/binarization technologies. Compared with multi-bit (2-8 bit) model quantization compressing parameters to integers, binarization usually directly applies the sign function to compress the model to a more compact 1-bit. Moreover, due to the application of binary parameters, bitwise operations (XNOR and popcount) can be applied to inference during actual deployment instead of integer multiply-add operations in 2-8 bit model quantization. Therefore, binarization is considered to take advantage of the hardware and can achieve more speedup than multi-bit quantization.

---

> > > > > ### Author Response · Authors · 2022-11-18
> > > > > **Response to Reviewer LQy6 (5/5)**
> > > > >
> > > > > >  **[Q7]** It is not clear to what extent the chosen datasets and tasks are challenging or representative. The paper should elaborate more on why some of the tasks have been chosen. The description in Section 3.1 assumes the reader knows the tasks but does not provide additional information on why they are representative.
> > > > >
> > > > > **[A7]** We further clarify in Appendix B of our revision why the selected tasks in BiBench are challenging and representative.
> > > > >
> > > > > To comprehensively evaluate the performance of the binarization algorithm in various learning tasks, we should select various representative tasks. First, representative perception modalities are selected in our deep learning, including (2D/3D) vision, text, and speech. Research on these modalities has progressed rapidly and has a broad impact, so we choose specific tasks and datasets in these modalities.
> > > > > Specifically, (1) on the 2D vision modality, we choose the basic image classification task and object detection task (one of the most popular downstream tasks), the former including CIFAR10 and ImageNet datasets, the latter including Pascal VOC and COCO datasets. These datasets (ImageNet and COCO) are more challenging large datasets, while CIFAR10 and Pascal VOC are more basic.
> > > > > For other modalities, binarization is still challenging even with the underlying tasks and datasets in the field since there are few related binarization studies: (2) In the 3D vision modality, the basic point cloud classification ModelNet40 dataset is selected to evaluate the binarization performance, which is regarded as one of the most fundamental tasks in 3D point cloud research and is widely studied. (3) In the text modality, the General Language Understanding Evaluation (GLUE) benchmark is usually recognized as the most popular dataset, including nine sentence- or sentence-pair language understanding tasks. (4) The keyword spotting task was chosen as the base task in the speech modality, specifically the Google Speech Commands classification dataset.
> > > > >
> > > > > Based on the above reasons and rules, we have selected a series of challenging and representative tasks for BiBench to evaluate binarization comprehensively and have obtained a series of reliable and informative conclusions and experiences.
> > > > >
> > > > > >  **[Q8]** Code: The implementation is not available. A benchmark should provide the code (in this case anonymous) for reproducibility. Moreover, the code has to be clear, well-documented, and easy to run.
> > > > >
> > > > > **[A8]** Thanks for pointing it out. We follow the reviewer's comments to release our first version code and corresponding documents in the supplementary material. We will improve this project continuously and publish it once upon acceptance.

---

> > > > > > ### Comment · Reviewer_LQy6 · 2022-11-22
> > > > > > **Thanks for the revision.**
> > > > > >
> > > > > > Thanks for the comprehensive revision in such a short time.
> > > > > >
> > > > > > The paper appears much more solid and easy to read. I have, though, a couple of remarks on the revision.
> > > > > >
> > > > > > - **Popcount**: Add an explanation in the paper
> > > > > > - **On image modality** → The text in the paper could be more explicit, about which tasks are image modality. The answer is clear, the paper only presents the word image modality but does not elaborate immediately. The explanation comes in Section 3.
> > > > > > - **Local structures**: The modified text about what local structures are should be in the main paper and not in the appendices.
> > > > > > - Severe accuracy loss: remove the capital letter
> > > > > >
> > > > > > The only major point on my side is still
> > > > > >
> > > > > > > As a benchmark, it should probably compare with other, simpler strategies to make the network more compact. For instance, one such strategy could be dropout or model quantization. There is no need to be exhaustive there, but there should be the possibility for researchers working on Network binarization to assess their methods on more traditional techniques.
> > > > > >
> > > > > > I understand the author's answer, but in short, I would have liked to see some algorithm included in the benchmark itself.
> > > > > >
> > > > > > For the above reasons (positive and negative), I raised my score.

---

> > > > > > > ### Author Response · Authors · 2022-11-29
> > > > > > > **Re: Thanks for the revision (1/2)**
> > > > > > >
> > > > > > > Thanks for your feedback and constructive suggestions! We are willing to provide detailed clarifications and improvements based on your 2nd-round comments. Since new revisions cannot be uploaded, specific changes are shown below in the form of "[location] content".
> > > > > > >
> > > > > > > >  **Q1**: Popcount: Add an explanation in the paper
> > > > > > >
> > > > > > > **A1**: Thank you for your suggestion, we revised the problem in the next version. Specifically, we clarified the definition of popcount below Eq.(1) of the manuscript, and the relevant content is revised as follows:
> > > > > > >
> > > > > > > [Section 2.1] $\operatorname{xnor}$ and $\operatorname{popcount}$ are bitwise instructions defined as [1,2], where the $\operatorname{popcount}$ instruction counts the number of bits with the "one" value in the input vector and writes the result to the targeted register.
> > > > > > >
> > > > > > > [1] Arm. Arm a64 instruction set architecture. https://www.amd.com/system/files/TechDocs/24594.pdf, 2020. Version: x.y.z.
> > > > > > >
> > > > > > > [2] AMD. Amd64 architecture programmer’s manual. https://developer.arm.com/documentation/ddi0596/2020-12/SIMD-FP-Instructions/CNT--Population-Count-per-byte-, 2022. Version: x.y.z.
> > > > > > >
> > > > > > > >  **Q2**: On image modality → The text in the paper could be more explicit, about which tasks are image modality. The answer is clear, the paper only presents the word image modality but does not elaborate immediately. The explanation comes in Section 3.
> > > > > > >
> > > > > > > **A2**: We revised the corresponding part of our manuscript according to the reviewer's suggestion, clarifying the mentioned concepts to make the text more explicit:
> > > > > > >
> > > > > > > [Section 1] However, since most binarization algorithm studies are engineered for learning tasks with 2D image inputs (described as 2D visual modality tasks, e.g., CIFAR10 and ImageNet classification tasks), their insights and conclusions are rarely verified in a broader range of tasks with other modality inputs, such as 3D point clouds, text, and speech.
> > > > > > >
> > > > > > > [Section 3] We selected 9 learning tasks with 4 different input modalities to comprehensively evaluate network binarization algorithms. For the widely-evaluated 2D visual modality tasks, ... For 3D visual modality tasks, ... For textual modality tasks, ... For speech modality tasks, ...
> > > > > > >
> > > > > > > >  **Q3**: Local structures: The modified text about what local structures are should be in the main paper and not in the appendices.
> > > > > > >
> > > > > > > **A3**: We listed the referred local structures in Section 2.1 of the last revision's main text. Here we followed your suggestion to further clarify the referred local structures and corresponding binarization algorithms:
> > > > > > >
> > > > > > > [Section 2.1] Note that for selected binarization algorithms, the techniques requiring specified local structures or training pipelines are excluded for a fair comparison, i.e., the bi-real shortcut of Bi-Real [1] and duplicate activation of ReActNet [2] in CNN neural architectures.
> > > > > > >
> > > > > > > [1] Liu Z, Shen Z, Savvides M, et al. Reactnet: Towards precise binary neural network with generalized activation functions. ECCV, 2020.
> > > > > > >
> > > > > > > [2] Liu Z, Wu B, Luo W, et al. Bi-real net: Enhancing the performance of 1-bit cnns with improved representational capability and advanced training algorithm. ECCV, 2018.
> > > > > > >
> > > > > > > >  **Q4**: Severe accuracy loss: remove the capital letter
> > > > > > >
> > > > > > > **A4**: Thanks for pointing it out. We corrected it.

---

> > > > > > > > ### Author Response · Authors · 2022-11-29
> > > > > > > > **Re: Thanks for the revision (2/2)**
> > > > > > > >
> > > > > > > > >  **Q5**: The only major point on my side is still: As a benchmark, it should probably compare with other, simpler strategies to make the network more compact. For instance, one such strategy could be dropout or model quantization. There is no need to be exhaustive there, but there should be the possibility for researchers working on Network binarization to assess their methods on more traditional techniques.
> > > > > > > > I understand the author's answer, but in short, I would have liked to see some algorithm included in the benchmark itself.
> > > > > > > >
> > > > > > > > **A5**: Thanks for your constructive feedback and suggestions! We further evaluated representative multi-bit quantization algorithms [1,2,3] (with INT2 and INT8) and dropout (pruning) algorithms [4,5] in the limited time to demonstrate their accuracy and efficiency metrics and compare them to network binarization. The results show that compared with multi-bit quantization and dropout, binarization brings more significant compression and acceleration while facing greater challenges from the decline in accuracy.
> > > > > > > >
> > > > > > > > To highlight the characteristics of network binarization, we compare it with other mainstream compression approaches, including multi-bit quantization and pruning, from accuracy and efficiency perspectives (Table A5). Overall, the results express the intuitive conclusion that there is a trade-off between accuracy and efficiency for different compression approaches. The ultra-low bit-width of network binarization brings acceleration and compression beyond multi-bit quantization and pruning. For example, binarization achieves 12.32x FLOPs saving while INT8 quantization just achieves 1.87x. However, binarization algorithms also introduce a significant performance drop, the largest among all compression approaches (e.g., CIFAR10-Res20 binary 85.74 vs. pruning 91.54). These results show that network binarization pursues a more radical efficiency improvement among existing compression approaches and is oriented for deployment on edge devices, which is consistent with the conclusions in BiBench.
> > > > > > > >
> > > > > > > > We hope these results address your concern and are willing to hear more suggestions.
> > > > > > > >
> > > > > > > >
> > > > > > > > Table A5. Accuracy and efficiency comparison among multi-bit quantization (2&8-bits), pruning, and binarization.
> > > > > > > >
> > > > > > > > |                   | Accuracy          |                   | Efficiency           |                      |
> > > > > > > > | ----------------- | ----------------- | ----------------- | -------------------- | -------------------- |
> > > > > > > > |                   | **CIFAR10-Res18** | **CIFAR10-Res20** | **FLOPs (Res20, G)** | **Param (Res20, K)** |
> > > > > > > > | FP32              | 94.82             | 91.99             | 13.61                | 11690                |
> > > > > > > > | **Binary (all)**  | **91.08 (-3.74)** | **85.74 (-6.25)** | **1.105 (12.32x)**   | **884 (13.22x)**     |
> > > > > > > > | DoReFa-INT2 [1]   | 92.71             | 87.56             | 1.686                | 1681                 |
> > > > > > > > | PACT-INT2 [2]     | 92.98             | 88.12             | 1.686                | 1681                 |
> > > > > > > > | LSQ-INT2 [3]      | 93.11             | 89.26             | 1.686                | 1681                 |
> > > > > > > > | **INT2 (all)**    | **92.93 (-1.89)** | **88.31 (-3.68)** | **1.686 (8.07x)**    | **1681 (6.95x)**     |
> > > > > > > > | DoReFa-INT8 [1]   | 94.79             | 91.83             | 7.278                | 4067                 |
> > > > > > > > | PACT-INT8 [2]     | 94.8              | 91.87             | 7.278                | 4067                 |
> > > > > > > > | LSQ-INT8 [3]      | 94.78             | 91.95             | 7.278                | 4067                 |
> > > > > > > > | **INT8 (all)**    | **94.79 (-0.03)** | **91.88 (-0.11)** | **7.278 (1.87x)**    | **4067 (2.87x)**     |
> > > > > > > > | Li et al. [4]     | 94.47             | 91.32             | 9.527                | 9936                 |
> > > > > > > > | ResRep [5]        | 94.53             | 91.76             | 6.805                | 7247                 |
> > > > > > > > | **Pruning (all)** | **94.50 (-0.32)** | **91.54 (-0.45)** | **8.166 (1.67x)**    | **8591 (1.36x)**     |
> > > > > > > >
> > > > > > > > [1] Zhou S, Wu Y, Ni Z, et al. Dorefa-net: Training low bitwidth convolutional neural networks with low bitwidth gradients[J]. arXiv preprint arXiv:1606.06160, 2016. (Citation 1773)
> > > > > > > >
> > > > > > > > [2] Choi J, Wang Z, Venkataramani S, et al. Pact: Parameterized clipping activation for quantized neural networks[J]. arXiv preprint arXiv:1805.06085, 2018. (Citation 583)
> > > > > > > >
> > > > > > > > [3] Esser S K, McKinstry J L, Bablani D, et al. Learned Step Size Quantization[C]//International Conference on Learning Representations, 2020. (Citation 326)
> > > > > > > >
> > > > > > > > [4] Li H, Kadav A, Durdanovic I,  et al. Pruning filters for efficient convnets[J]. arXiv preprint arXiv:1608.08710, 2016. (Citation 3079)
> > > > > > > >
> > > > > > > > [5] Ding X, Hao T, Tan J, et al. ResRep: Lossless CNN Pruning via Decoupling Remembering and Forgetting[C]//International Conference on Computer Vision, 2021. (Citation 46)

---

### Author Response · Authors · 2022-11-18
**General Response**

We are grateful for all reviewers' feedback toward BiBench. To assist a clearer understanding of our paper, we summarize our main contributions below:

Our BiBench is a carefully engineered benchmark with in-depth analysis for network binarization and the first to inspect the requirements of binarization in the actual production setting. We define the comprehensive evaluation tracks and corresponding metrics for fairness and systematicness, i.e., learning task, neural architecture, and corruption robustness for accurate binarization, training consumption, theoretical complexity, and hardware inference for efficient binarization. We perform an exhaustive evaluation with various milestone binarization algorithms and suggest establishing a paradigm for accurate and efficient binarization among existing techniques. Our benchmark is valuable and meaningful for the ML community and will be a foundation for studying model binarization in broader and more practical scenarios.

We also updated our manuscripts; the main changes we made include the following:

(1) We carefully revised the whole manuscript to address the issues mentioned by reviewers, and other typos, broken sentences, and bad use of English.

(2) We clarified our manuscript's related concepts and claims in detail, including the details of binarization algorithms (Section 1, Section 2.1, and Appendix A), metric definition (Section 3.1 and Appendix B), implementation (Section 4 and Appendix A), colors in tables (Section 5), etc.

(3) We clarified the selection rules for binarization algorithms and learning tasks and why they are representative in Appendix A and B.

(4) We compared network binarization with other compression technologies in Section 1 and Appendix A.

(5) We open-sourced the BiBench code and related documentation in the supplementary material.

For detailed explanations, please see our responses to each reviewer.

---

### Decision · Program_Chairs · 2023-01-20

**Decision:**

Reject

**Justification For Why Not Higher Score:**

N/A

**Justification For Why Not Lower Score:**

N/A

**Metareview: Summary, Strengths And Weaknesses:**

The paper presents a comprehensive study of network binarization, with the goal of establishing a systematic benchmark for accurate and efficient binarization techniques. To achieve this, the authors propose several evaluation tracks and corresponding metrics. These tracks include tasks, neural architecture choices, robustness,  training consumption, theoretical complexity, and hardware inference costs.

The paper has some limitations that should be addressed. One such limitation is the selection of BNNs for evaluation, which includes a mix of older and newer BNNs but lacks a clear and convincing rationale for their inclusion (given that there are over 2000 papers proposing BNNs, why should we care about those specific ones?). Additionally, the choice of optimization methods is limited, which could affect the validity of the results and their comparability to more recent and future methods. Furthermore, the selection of BNNs does not always lead to clear and definitive conclusions, as can be seen in Table 3.

Additionally, while the main comparison in the paper is in typically expressed in relative performance to full precision, other and perhaps more informative metrics in terms of trade-offs might be considered. For instance, comparing the accuracy of different BNNs against the number of bits used, the latency against accuracy, and the training efforts against accuracy could provide valuable insights into the strengths and weaknesses of the different methods.

We encourage the authors to  address the wealth of comments raised by the reviewers and the above limitations and believe that revised version of the paper can be a valuable contribution to the field.